# Interleukin-35 impairs human NK cell effector functions and induces their ILC1-like conversion with tissue residency features

Valentin Picant [1,6], Lara Revol-Bauz[1,6], Laurie Tonon [2], Timothée Casini[1], Aurélien Voissière [1], Dominique Poujol[1], Emilie Picard[1], Céline Rodriguez[1,3], Cyril Degletagne[4], Emily Sible[5], Uzma Hasan [3,5], Anthony Ferrari[2], Christophe Caux[1,3] & Nathalie Bendriss-Vermare [1,3] ✉

Natural Killer (NK) cells play pivotal immunological roles including direct cytotoxic effector function and secretion of inflammatory and immunomodulating cytokines. In the context of chronic inflammation, NK cell fitness decreases during disease progression through currently unknown mechanisms. Here, we demonstrate that Interleukin-35 (IL-35) inhibits human NK cell proliferation, pro-inflammatory, and cytotoxic functions, while promoting secretion of TGF-β and proangiogenic factors in vitro. We show prolonged exposure to IL-35 converts both conventional and adaptive NK cells into CD9+CD103+CD49a+ ILC1-like cells via autocrine TGF-β. We assess cancer patient-derived public datasets and reveal the presence of IL-35-producing cells and IL-35-receptor-expressing NK/ILC1-like cells within the tumor microenvironment and associate IL-35 with poor prognosis. Collectively, our findings identify and implicate IL-35 as a key driver of NK cell plasticity, promoting the acquisition of features associated with tissue residency and weakened effector functions, and could be relevant in pathophysiological contexts, highlighting IL-35 as an attractive target for future immunotherapies aimed at enhancing NK cell clinical activity.

Natural Killer cells (NK) are part of the Innate Lymphoid Cells (ILC) family along with tissue-resident ILC1, ILC2, ILC3, and Lymphoid Tissue inducer[1]. They are key players against infections and cancers[2,3] through cytolytic activities against dangerous cell and secretion of cytokines (IFN-γ, TNF-α, FLT3L, GM-CSF) and chemokines (XCL1 and CCL3/4/5) that can modulate the attraction, survival, and function of other immune cells[4]. NK cells are finely regulated by a balance between activating and inhibitory external signals, among which cytokines such as interleukin (IL)−2, IL-12, IL-15, and IL-18 promote NK cell activation

and survival[3]. On the contrary, hypoxic and/or immunosuppressive environment enriched in inhibitory cytokines (i.e,. TGF-β), as observed during chronic infections or cancer, progressively dampen NK cell effector functions[5,6], while triggering their ability to secrete pro-angiogenic factors[7,8]. Human mature NK cells can be subdivided into three distinct subpopulations known as conventional (ConvNK) CD56Bright, ConvNK CD56DimNKG2Cneg and adaptive (AdaptNK) CD56DimNKG2Cpos NK cells emerging after cytomegalovirus (CMV) infections[9,10]. CD56Bright ConvNK are the main source of XCL1/2 and GM-

[1]CISTAR team, Cancer Research Center of Lyon, INSERM U1052, CNRS UMR5286, Université de Lyon, Université Lyon 1, Centre Léon Bérard, Lyon, France. [2]Synergie-Lyon-Cancer Fondation, Centre Léon Bérard, Lyon, France. [3]Lyon Immunotherapy for Cancer Laboratory (LICL), Centre Léon Bérard, Lyon, France. [4]Genomic platform, Centre de Recherche en Cancérologie de Lyon, INSERM U1052, CNRS UMR5286, Université de Lyon, Université Lyon 1, Centre Léon Bérard, Lyon, France. [5]Centre International de Recherche en Infectiologie, CIRI, INSERM U1111, Lyon, France. [6]These authors contributed equally: Valentin Picant, Lara Revol-Bauz. ✉e-mail: nathalie.bendriss-vermare@lyon.unicancer.fr

CSF and present the highest proliferative potential while all CD56[Dim] ConvNK cells are professionalized in CCL3/4/5 secretion and have higher cytotoxic properties[11–13]. In addition, AdaptNK present adaptive-like features with increased effector functions against CMV-infected cells[14]. The recently described TGF-β-mediated conversion of NK cells into CD49a+CD103+CD9+ ILC1-like cells revealed a previously under-appreciated inter-ILCs plasticity and NK/ILC1 family heterogeneity[15–17]. As an intermediate state between NK and ILC1, NK-derived ILC1-like cells have been described in several chronic inflammations[17,18], including cancers[15,19–22]. Functionally, NK-derived ILC1-like cells display a reduced ability to kill target cells and to interact with anti-tumor immune cells while they have a greater ability to secrete pro-angiogenic factors. They have been described to participate to disease pathogenesis in mouse models[15,18,23–25] and are associated with poor outcome in cancer patients[21,22]. Yet, the contexts and mechanisms driving the emergence of NK-derived ILC1-like cells in humans remain poorly understood.

IL-35 is a member of the IL-12 family, existing as a heterodimer of IL-12p35 and EBI3[26], with immunosuppressive activity. IL-35 is secreted by a large spectrum of regulatory cells such as Tregs[27,28], Bregs[29,30], tolerogenic/suppressive myeloid cells[31–33] and trophoblasts[34]. Invariably, IL-35 has been linked with disease pathogenesis and poor survival in mouse models of chronic inflammatory diseases or cancers and similar correlations were made in patients[35–39]. The IL-35 receptor may be a heterodimer or homodimer consisting of the GP130, IL-12Rβ2 chain, and/or IL-27Rα, which activates signaling pathways dependent on the STAT proteins. Potential configurations of IL-35 receptors are GP130-GP130, IL-12Rβ2-IL-12Rβ2, IL-12Rβ2-GP130, and IL-12Rβ2-IL-27Rα[40]. IL-12Rβ2 is expressed on the surface of activated T cells and NK cells, IL-27Rα is mainly expressed by activated CD8+ T cells, CD4+ T cells, B cells, monocytes, whereas GP130 is expressed by most immune cells[26]. IL-35 exerts an immunosuppressive effect by inhibiting B/T cells, dendritic cells, and type 1 macrophages[27–29,41–44] and by promoting, expanding, and activating immunoregulatory cells[28,30,31,34,42,45–47]. Furthermore IL-35 also facilitates tumor angiogenesis[48,49]. However, its effects on primary NK cells have never been characterized.

In this study, we evaluated in vitro the role of IL-35 in NK cell biology through short- and long-term exposure of human primary blood NK cells to IL-35, thus mimicking acute and chronic pathologies in which NK cell dysfunction occurs. Our results highlight strong suppressor effects of IL-35 on NK cell proliferation, pro-inflammatory, and cytotoxic functions, while promoting their secretion of TGF-β and proangiogenic factors. We also describe the conversion of both ConvNK and AdaptNK into ILC1-like cells with tissue residency features and low effector function in response to long-term exposure to IL-35. This conversion was dependent on autocrine TGF-β. Cancer patient-derived public data analysis revealed the presence of IL-35-producing cells and IL-35-receptor-expressing NK/ILC1-like cells within tumor microenvironment and associated IL-35 with poor prognosis. Our findings identify the immunosuppressive factor IL-35 as a key driver of NK cell plasticity towards the acquisition of tissue residency features with weakened effector functions. This might be of importance in chronic inflammatory diseases and provides a basis for future IL-35-targeting therapies.

## Results

### IL-35 inhibits human NK cell activation and polarizes their secretory profile

We first evaluated the impact of IL-35 on human NK cell activation and IFN-γ production in response to activating cytokines (IL-12, IL-15, IL-2, IL-18)[3]. We observed that cytokine-induced IFN-γ secretion and CD25 expression in NK cells were strongly reduced in the presence of IL-35 in every condition (Fig. 1a, Supplementary Fig. 1a, b), without impacting NK cells viability (Supplementary Fig. 1c). The strongest inhibition was

observed for IL-12 and IL-18 combination (Supplementary Fig. 1a, b), and appeared to be dose-dependent (Supplementary Fig. 1d, e). A dose response experiment helped us to select the IL-35 concentration of 100 ng/ml for the rest of the experiments, as commonly used in other studies[29].

We then measured the acquisition of the activation markers 4-1BB, PD-L1, LAG3, and CD69, in addition to IFN-γ and CD25, following IL-12 + IL-18 activation in presence or absence of IL-35. The frequency of positive NK cells was strongly reduced in the presence of IL-35 for each individual marker except CD69 (Fig. 1b). T-sne analysis allowed the identification of a highly activated NK cell cluster concomitantly expressing all 6 markers at high intensities (Fig. 1c and Supplementary Fig. 1f). We observed a significant reduction of the proportion of IL-12 + IL-18-induced highly activated NK cells in the presence of IL-35 (Fig. 1c). Beside IFN-γ, IL-35 induced a global impairment of pro-inflammatory cytokines and chemokines production by IL-12 + IL-18- and IL-15 + IL-18-activated NK cells, with a significant decrease (-30–70%) of TNF-α, GM-CSF, CCL3, CCL4, XCL1, and FLT3L (Fig. 1d and Supplementary Fig. 1g). In accordance, reduced expression of T-BET and EOMES in the presence of IL-35 also illustrates its negative impact on NK cell effector functions (Fig. 1e and Supplementary Fig. 1h)[50]. Interestingly, IL-35-exposed NK cells acquired the ability to secrete higher levels of pro-angiogenic (VEGF-A, IL-8, CCL5) factors (Fig. 1f).

We next evaluated in human NK cells the expression of GP130, IL-12Rβ2, and IL-27Rα that have been described to compose IL-35 receptor (IL-35R) on the surface of T cells and B cells mainly[26]. The GP130 chain was not expressed in resting NK cells but its expression was specifically induced by IL-12 in 40% of NK cells and the addition of IL-15 or IL-18 did not change the expression levels (Fig. 1g). Very few NK cells (5–10%) constitutively expressed IL-12Rβ2, while IL-27Rα was expressed in 35–45% of total NK cells. IL-12Rβ2 was upregulated in response to individual cytokines and to a greater extent when cytokines were combined. IL-27Rα was strongly upregulated in NK cells in response to IL-2, IL-15 and IL-18 and to a lesser extent to IL-12 to reach >90% IL-27Rα positive NK cells in response to cytokine combination (Fig. 1g).

Our results indicate that human NK cells are a direct target of IL-35, which decreases their activation and pro-inflammatory functions while promoting a pro-angiogenic profile.

### IL-35 impairs human NK cell proliferation but not survival

We next investigated the ability of human NK cells, preactivated for 16 h with cytokines known to induce the IL-2 receptor CD25 (i.e., IL-12 +/− IL-15), to proliferate in response to IL-2 in the presence of IL-35 for 3, 5, 7, and 9 days (Fig. 2a and Supplementary Fig. 2a). Analysis of CellTrace Violet (CTV) dilution showed a time-dependent NK cell proliferation in response to IL-2, which was gradually enhanced from medium -> IL-12 -> IL-15 -> IL-12 + IL-15 pre-activation conditions (Fig. 2b), coherently with CD25 expression observed at the end of the priming phase (Supplementary Fig. 2a). Invariably, regardless the expression level of CD25, IL-35 significantly reduced NK cell proliferation in response to IL-2 (Fig. 2b, c), in agreement with macroscopic observation of culture wells (Supplementary Fig. 2b), and with a lower cell number (Fig. 2d) compared to IL-2 alone. Importantly, NK cell survival was similar in the presence or absence of IL-35 (Fig. 2e) even at day 9, indicating that the decreased cell number was not due to cell death. These data demonstrate that IL-35 is also a strong inhibitor of human NK cell proliferation without impacting IL-2-induced pro-survival effects.

### Long-term exposure to IL-35 leads to human NK cell hypo-responsiveness and altered NK cell surface receptor expression

To mimic the impact of IL-35 long-term exposure on NK cell functions and phenotype as observed in chronic inflammations, human NK cells

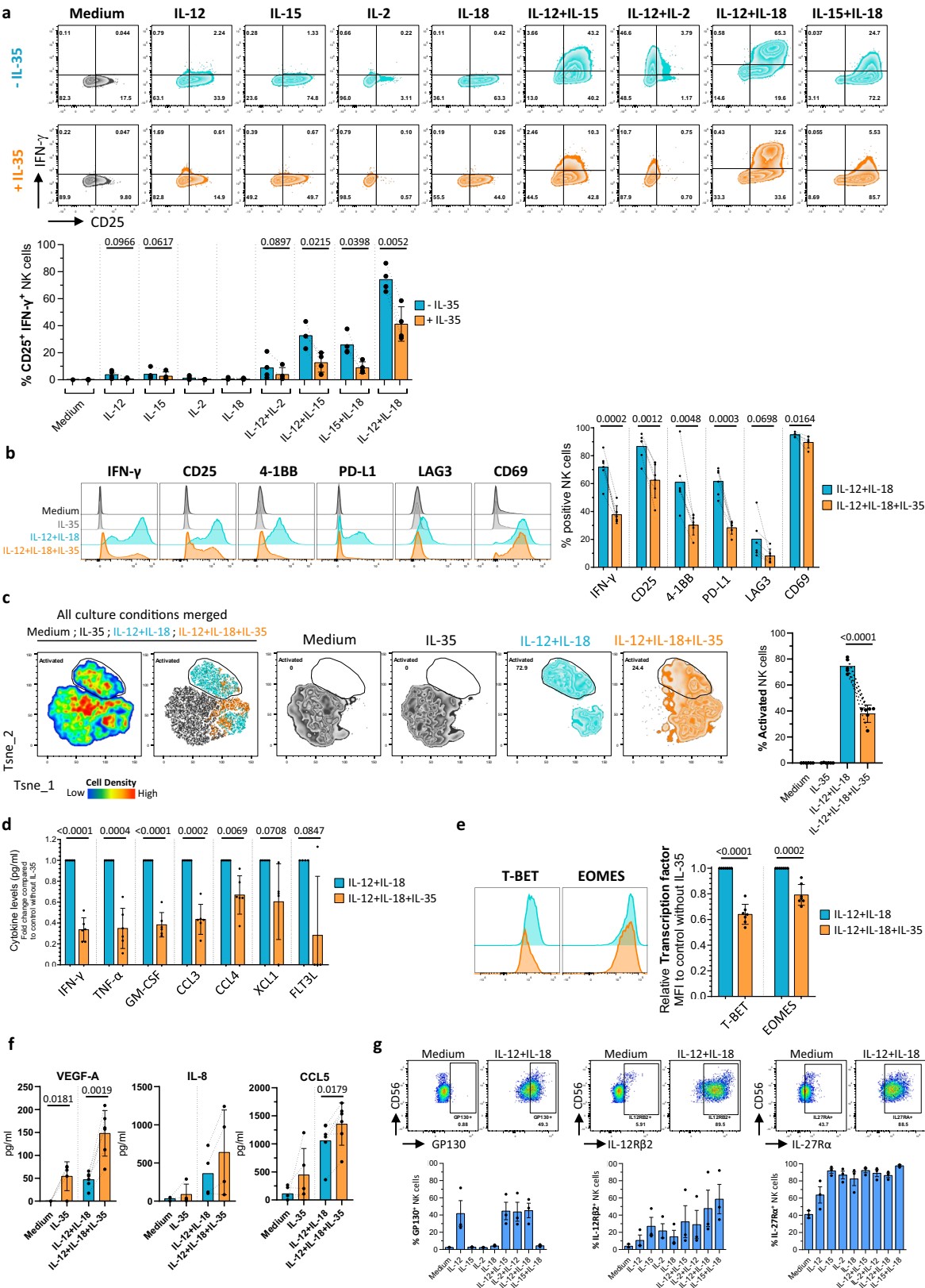

were cultured in IL-2 vs IL-2 + IL-35 for 7 days and then reactivated with IL-12 + IL-18 for 16 h or with K562 cells for 4 h to assess their capacity to produce pro-inflammatory cytokines and chemokines or to kill target cells, respectively (Fig. 3a). As shown in Fig. 3b, the frequency of IFN-γ-producing cells as well as the intensity of IFN-γ expression following IL-12 + IL-18 activation were strongly reduced in IL-35-exposed NK cells.

This observation was confirmed following quantification of IFN-γ levels in NK cell culture supernatants and extended to other cytokines and chemokines (TNF-α, GM-CSF, CCL3/4, XCL1, and FLT3L) (Fig. 3c and Supplementary Fig. 3a). In addition, IL-35 strongly decreased NK cell cytotoxicity towards K562 target cells (Fig. 3d). In line with their reduced killing functions, we found that NK cells cultured with IL-

**Fig. 1 | Short-term exposure to IL-35 inhibits human NK cell activation and secretion of pro-inflammatory cytokines/chemokines. a** NK cells were cultured for 24 h with IL-12, IL-15, and IL-18 (10 ng/mL), IL-2 (1000 UI/ml), and IL-35 (100 ng/ml) or their combination. Representative flow cytometry plots for CD25 and IFN-γ expression (upper) and quantification (%) (lower) of CD25⁺ IFN-γ⁺ NK cells. Mean values ± S.D are shown (*n* = 4 individual donors). **b** Representative raw histograms of IFN-γ, CD25, 4-1BB, PD-L1, LAG3, and CD69 in NK cells (left) and cumulative histogram of % positive NK cells in different culture conditions (right). Mean values ± S.D are shown (*n* = 5 to 6 individual donors). **c** Two-dimensional T-sne plots showing NK cell clustering based on surface marker profiles (IFN-γ/CD25/4-1BB/PD-L1/LAG3/CD69), after culture as in Fig. 1b. Representative T-sne (left) and quantification (%) (right) of activated NK cells. Mean values ± S.D are shown (*n* = 6 individual donors). **d** Cytokines and chemokines were quantified by ECLIA assay in

supernatants from NK cells cultured for 24 h in the presence of IL-12 and IL-18 with or without IL-35. Results are expressed as fold change relative to control without IL-35. Mean values ± S.D are shown (*n* = 4 to 6 individual donors). **e** Representative flow cytometry plots (left) and quantification (right) of T-BET and EOMES expression in NK cells cultured as in Fig. 1d. Results are expressed as MFI expression in IL-35 condition relative to control without IL-35. Mean values ± S.D are shown (*n* = 7 individual donors). **f** VEGF-A, IL-8, and CCL5 were quantified by ECLIA assay in supernatants from NK cells cultured for 24 h as in Fig. 1b. Mean values ± S.D are shown (*n* = 3 to 6 individual donors). **g** Representative raw histograms (upper) and cumulative histograms (lower) of % positive NK cells for GP130, IL-12Rβ2, IL-27Rα after culture as indicated for 24 h. Mean values ± S.D are shown (*n* = 3 individual donors). Statistical significance was determined using paired T test. Source data are provided as a Source Data file.

2 + IL-35 for 7 days expressed lower levels of the activating receptors NKp30, NKp44, NKp46, DNAM1, and NKG2D than control IL-2-cultured NK cells (Fig. 3e, f and Supplementary Fig. 3b, c). Furthermore, IL-35 unexpectedly downregulated the expression of the inhibitory receptors NKG2A and TIGIT, while CD96 and LAG3 were slightly upregulated and no significant difference was observed for PD-L1 and TIM3 (Fig. 3g, h and Supplementary Fig. 3d, e). Using the combined expression of activating receptors (Fig. 3f and Supplementary Fig. 3c) but not inhibiting receptors (Fig. 3h and Supplementary Fig. 3e), T-sne analysis identified two distinct NK cell clusters clearly differentiating IL-2 and IL-2 + IL-35 conditions. These results indicate that NK cells, chronically exposed to IL-35, display a dysfunctional/hyporeactive state that is associated with an altered expression of NK cell surface receptors.

## scRNA-seq reveals transcriptional regulation and mechanisms involved in NK cell subsets' response to IL-35

For a deep analysis of the effects of IL-35 on NK cells, we performed scRNA-seq analysis of human NK cells exposed to IL-2 or IL-2 + IL-35 for 2 and 4 days. Unsupervised clustering identified 9 clusters (C) (Fig. 4a) consisting of CD56^Dim NK cells (C1, C3, C8, C9), CD56^Bright NK cells (C6), and adaptive NK cells (C4, later referred to as AdaptNK), as confirmed by their respective specific marker genes (Supplementary Fig. 4a) and by using CD56^Bright and CD56^Dim gene signatures (Supplementary Fig. 4b, c and Table 1). The clusters 5 and 7 and part of clusters 2 and 6 were associated with cell proliferation (Fig. 4b). Interestingly, only cluster 3 was found to be strictly dependent on the presence of IL-35 (Fig. 4c). We then analyzed the impact of IL-35 on AdaptNK, CD56^Dim and CD56^Bright NK cells altogether referred to as ConvNK, and proliferative clusters (ProlifNK) comparing pathways enrichment (Fig. 4d, e) and differentially expressed genes (DEGs) (Fig. 4f, g) in the presence versus absence of IL-35. In accordance with our previous data (Figs. 1, 3, 4), gene set-enrichment analyses highlighted a significant decrease of pathways associated with NK cell activation and effector functions (Fig. 4d) with a consistent downregulation of genes associated with NK cell cytotoxicity (*i.e. GZMA/K/B, GNLY, PRF1*), pro-inflammatory cytokines (*i.e. IFNG, CCL3/4, XCL1/2, CSF2*), and activating receptors (*i.e. TNFRSF9, NCR3, CD244, CD226, FCGR3A, SLAMF7, KLRB1, KIR2DL4*) (Fig. 4f) as well as the overexpression of genes associated with reduced functionality (*i.e. CD96, CISH, SOCS1, SH2D1B*) (Fig. 4g), in favor of tissue migration and residency (*i.e. ITGA1, ITGAE, CD9, CXCR3/4, RUNX2*), TGF-β response and signaling (*i.e. SKIL, SKI, SOCS1, SOX4, ZFHX3, CLIC3, SMAD7, ACVR1B*), and the induction of angiogenesis (*i.e. CCL5, TGFB1, GIT1*) (Fig. 4e–g). This general IL-35-induced transcriptomic switch is consistent across all three NK populations (Fig. 4d, e), coherently with the similar expression and regulation of IL-35R chains in conventional NKG2C⁻ and adaptive NKG2C⁺ NK cells (Fig. 4h), the decreased production of IFN-γ (Fig. 4i), and the increased production of VEGF-A (Fig. 4j) in the presence of IL-35. Nevertheless, individual DEGs appeared to be different within each NK population

with only 15.9% of downregulated and 26.4% of upregulated genes being shared while the majority of DEGs were observed in only one NK population (Fig. 4f, g). We then separated AdaptNK, ConvNK, and ProlifNK (Supplementary Fig. 4d), and observed a significant increase in an ILC1-like gene signature score (Table 1) in the presence of IL-35 for all three populations (Supplementary Fig. 4e). Unsupervised clustering revealed the existence of an IL-35-induced cluster into AdaptNK (C3) and ConvNK (C2), while all clusters observed in ProlifNK were mostly enriched in IL-2-treated NK cells, consistently with the anti-proliferative effect of IL-35 (Fig. 4k, l and Supplementary Fig. 4e). The ILC1-like gene signature was significantly enriched in these IL-35-induced AdaptNK and ConvNK clusters (Fig. 4m and Supplementary Fig. 4e). The comparison of the genes specifically expressed in the identified ILC1-like cluster versus all other clusters revealed only a partial overlap between marker genes of AdaptNK- and ConvNK-derived ILC1 cells (Fig. 4n). Our results highlight a major transcriptional regulation of all human NK cell subsets by IL-35 leading to loss of effector functions and acquisition of a proangiogenic/TGF-β profile.

## IL-35 drives human NK cell conversion into an irreversible ILC1-like phenotype

It was recently shown that NK cells can convert into ILC1-like cells with reduced cytotoxicity[15,16]. As we identified an ILC1-like signature in NK cells exposed to IL-35 by scRNA-seq, we evaluated the potential of IL-35 to induce human NK cell conversion into ILC1-like cells over the course of an 8-day culture. Long-term exposure of human NK cells to IL-35 upregulated the expression of tissue residency markers such as CD49a, CD103, and CD9 (Supplementary Fig. 5a) with the emergence of a specific cluster in T-sne analysis (Supplementary Fig. 5b), in a time-dependent manner (Fig. 5a and Supplementary Fig. 5c). At day 8 in presence of IL-35 nearly all cells adopt a converted ILC1-like cell phenotype while IL-2 alone did not promote this conversion (Fig. 5a and Supplementary Fig. 5d). Chronic exposure to IL-35 also lowered the expression of EOMES in human NK cells compared to IL-2 alone (Fig. 5b) while the level of T-BET expression did not change (Fig. 5c). Although both cycling and non-cycling cells acquired an ILC1-like profile, the highest levels of CD9, CD103, and CD49a were observed on cycling cells (Fig. 5d). To assess whether IL-35-induced effects on NK cells were reversible, we cultured NK cells in the presence of IL-35 to induce their ILC1-like conversion and hyporesponsiveness and then starved them or not of IL-35 for 2 additional days before activation with IL-12 and IL-18 for 16 h (Supplementary Fig. 5e). IFN-γ secretion by IL-35-starved NK cells remained impaired and similar to NK cells that were not starved of IL-35 (Fig. 5e). Consistently, the expression of ILC1-like markers continued to increase after IL-35 removal (Fig. 5f). Interestingly, hyporesponsiveness of long-term TGF-β-exposed NK cells appeared, in contrast, to be partially reversible in similar conditions (Supplementary Fig. 6a-b-c). Altogether these results demonstrate that IL-35 induces a conversion of NK cells into an irreversible ILC1-like phenotype with tissue residency features.

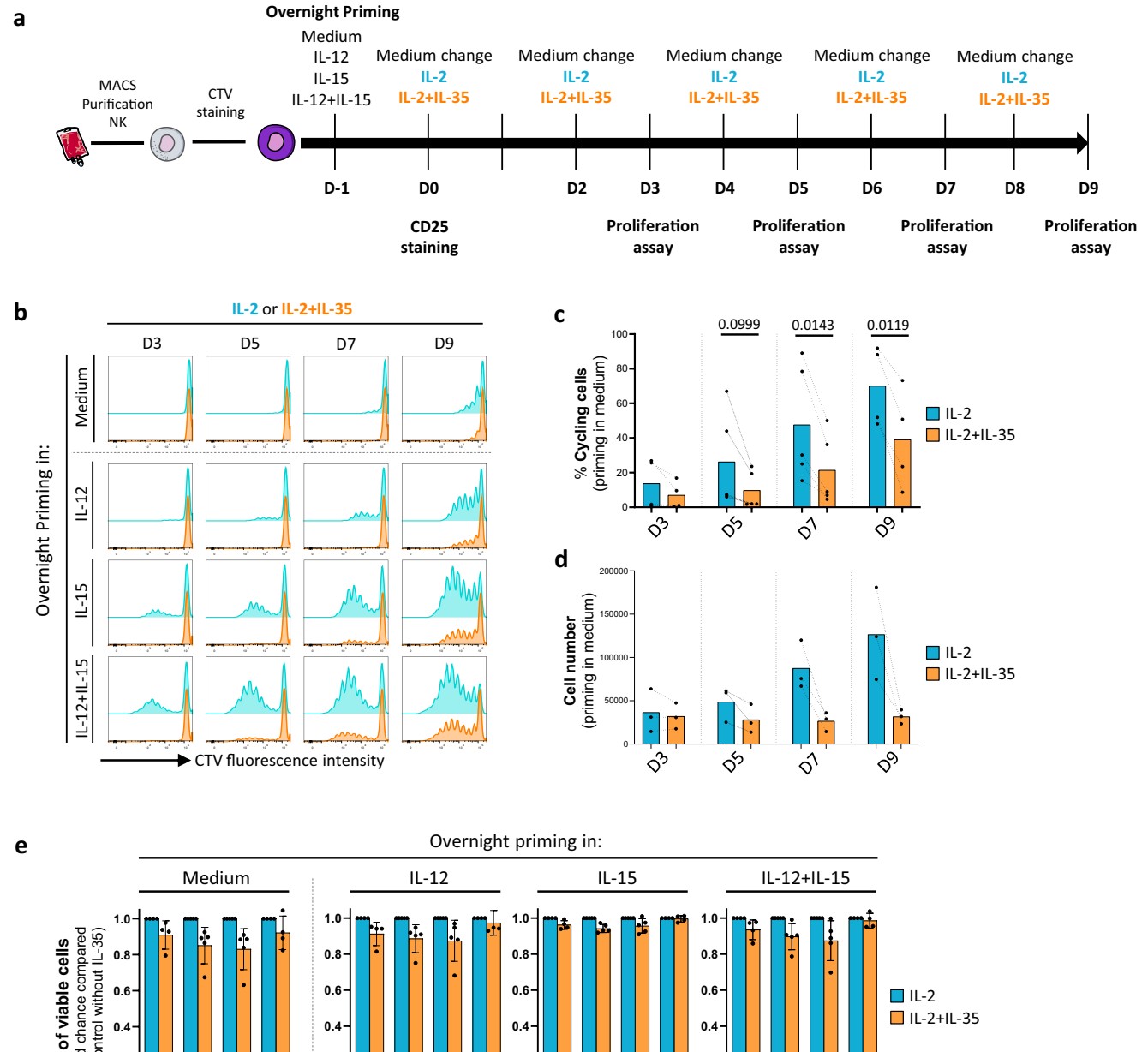

**Fig. 2 | IL-35 inhibits human NK cell proliferation but not survival. a** Schematic representation of the experimental protocol. NK cells were labeled with CTV at day 0 (D0), pre-activated for 16 h as indicated, and cultured in low dose IL-2 (100 UI/mL) with or without IL-35 (100 ng/ml) for 3, 5, 7, and 9 days (D3, D5, D7, D9, respectively). Images were provided by Servier Medical Art (https://smart.servier.com/), licensed under CC BY 4.0 (https://creativecommons.org/licenses/by/4.0/). **b** Representative flow cytometry plots for CTV fluorescence intensity for all pre-activation conditions and **c, d** quantification of % cycling cells and number of cells

that were primed in medium and after 3, 5, 7, and 9 days of proliferation in IL-2 versus IL-2 + IL-15. Mean values ± S.D are shown (*n* = 4 to 5, and 3 individual donors, respectively). **e** Quantification of the proportion of viable NK cells for all pre-activation conditions after 3, 5, 7 and 9 days of proliferation in IL-2 with or without IL-35. Results are expressed as fold change compared to the IL-2 control condition without IL-35. Mean values ± S.D are shown (*n* = 3 to 5 individual donors). Statistical significance was determined using paired T test. Source data are provided as a Source Data file.

## IL-35-triggered autocrine TGF-β drives NK cell dysfunction and conversion into ILC1-like cells

TGF-β is a prototypic inhibitor of NK cells[20,51], recently reported to drive NK-to-ILC1-like cell conversion[15,16] and we observed TGF-β-related pathways in NK cells exposed to IL-35 by scRNA-seq (Fig. 4e). Altogether these observations prompted us to evaluate whether TGF-β signaling was required for IL-35-triggered human NK cell conversion into ILC1-like cells. We observed that IL-35 upregulated the expression

of *TGFB1* in total NK cells compared to IL-2 alone (Fig. 6a) and that IL-35-exposed NK cells acquired the ability to secrete active TGF-β1 (Fig. 6b), in both conventional and adaptive subsets (Supplementary Fig. 6d). We used an anti-TGFβ1/2/3 neutralizing antibody or Galunisertib, a clinical-stage specific TGF-β-receptor inhibitor[52], that we validated as efficient inhibitors of the TGF-β-induced NK cell conversion into ILC1-like cells (Supplementary Fig. 6e), to block TGF-β signaling in NK cell cultures. We showed that blocking autocrine TGF-β1 restored NK cell proliferation

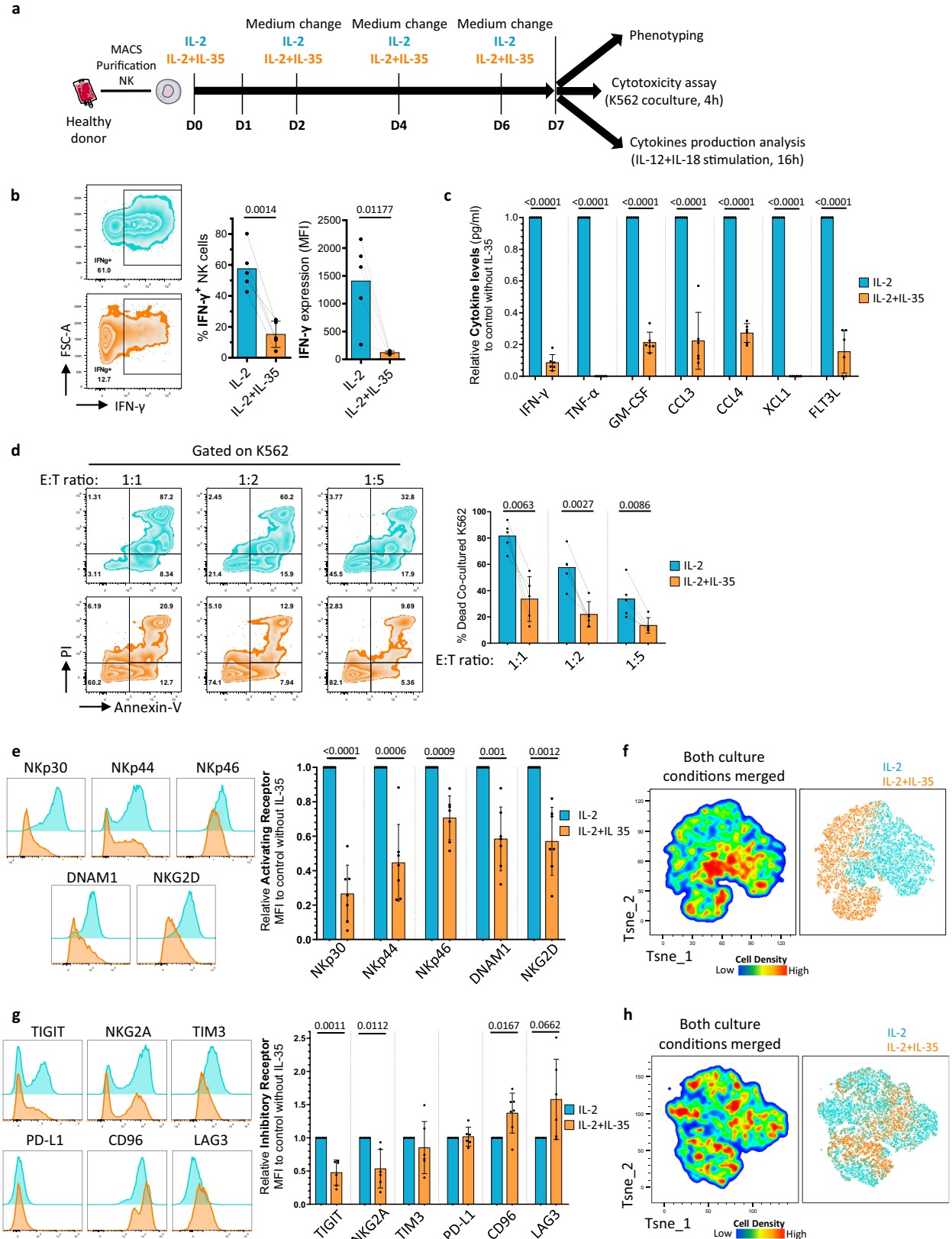

(Fig. 6c), prevented the downregulation of EOMES, T-BET, and IFN-γ expression in short-term (Fig. 6d) and long-term (Fig. 6e and Supplementary Fig. 6f) settings, and prevented the acquisition of tissue residency markers in presence of IL-35 (Fig. 6f). These results demonstrate that endogenous TGF-β signaling is required in IL-35-triggered human NK cell conversion into ILC1-like cells.

## NK/ILC1-like cells expressing IL-35R are present in tumors and IL-35 is associated with poor prognosis

To evaluate whether our observations might be relevant in pathophysiological contexts, we collected data from four different scRNA-seq datasets including Lung (GSE131907[53]), Esophagus (GSE160269[54]), Colon (GSE166555[55]) and Breast (GSE176078[56]) cancers from 108

**Fig. 3 | Long term exposure to IL-35 leads to NK cell hyporesponsiveness and altered NK cell surface receptor expression. a** Schematic representation of the experimental protocol. NK cells were cultured in low dose IL-2 (100 UI/mL) with or without IL-35 (100 ng/ml) for 7 days and then used for phenotyping analysis, cocultured with K562 at 1:1, 1:2, 1:5 E/T ratio for 4 h (cytotoxicity assay) or with IL-12 and IL-18 for 16 h (cytokine production analysis). Images were provided by Servier Medical Art (https://smart.servier.com/), licensed under CC BY 4.0 (https://creativecommons.org/licenses/by/4.0/). **b** Representative flow cytometry plots for IFN-γ expression (left) and quantification of IFN-γ production (% and MFI) (right) after 7 days in IL-2 with or without IL-35 followed by IL-12 and IL-18 for 16 h. Mean values ± S.D are shown (*n* = 5 individual donors). **c** Cytokines and chemokines were quantified by ECLIA assay in NK supernatants after 7d-culture in IL-2 with or without IL-35 followed by IL-12 and IL-18 for 16 h. Results are expressed as fold change in IL-35 compared to control without IL-35. Mean values ± S.D are shown (*n* = 6 individual donors). **d** Representative flow cytometry plots for AnnexinV/PI expression (left)

and quantification of % dead K562 cells (right) after 7d-culture of NK cells in IL-2 with or without IL-35 followed by 4h-coculture NK/K562. Mean values ± S.D are shown (*n* = 5 individual donors). **e** Representative flow cytometry plots (left) and quantification of activating receptors' expression (right) after 7d-culture, as indicated in Fig. 3a. Results are expressed as MFI in IL-35 relative to control without IL-35. Mean values ± S.D are shown (*n* = 7 individual donors). **f** Representative two-dimensional T-sne plots showing NK cell clustering based on NKp30, NKp44, NKp46, DNAM1, NKG2D expression (*n* = 7 individual donors). **g** Representative flow cytometry plots (left) and quantification of inhibitory receptors' expression (right) after 7d-culture, as indicated in Fig. 3a. Results are expressed as MFI in IL-35 relative to control without IL-35. Mean values ± S.D are shown (*n* = 6 to 7 individual donors). **h** Representative two-dimensional T-sne plots showing NK cell clustering based on NKG2A, TIGIT, TIM3, PD-L1, CD96, LAG3 expression (*n* = 6 to 7 individual donors). Statistical significance was determined using paired T test. Source data are provided as a Source Data file.

patient samples (Supplementary Fig. 7a). Following quality control and integration to correct batch effect (Supplementary Fig. 7b), a total number of 354,379 cells were included in this cancer atlas (Fig. 7a). Within this harmonized dataset, cells were annotated on the basis of canonical gene score signatures (Table 2). A homogenous repartition of cell populations was found between each dataset (Fig. 7a and supplementary Fig. 7c). The largest cluster consists of epithelial cells, predominantly composed of cancer cells. In addition, various immune-cell populations formed separate clusters, including T cells (conventional (Tconv) and regulatory (Treg)), NK(T) cells, B cells, plasma cells, monocytes, macrophages, dendritic cells (DC) and mast cells. Finally, we identified clusters of non-immune cells (endothelial cells, perivascular cells, and fibroblasts) and clusters of proliferating cells. We then performed in silico analysis to profile the expression of IL-35 (*EBI3* and *IL12A*) and its receptors (*IL6ST*, *IL12RB2*, and *IL27RA*) in tumors. *EBI3* was mostly expressed in DC and Treg and at lower levels in Tconv, monocytes/macrophages, and B cells (Fig. 7b and Supplementary Fig. 7d). *IL12A* was hardly detected due to drop-out with the highest expression levels observed in B and plasma cells, followed by conventional T cells (Fig. 7b and Supplementary Fig. 7d). The joint density map nicely highlights co-expression of *EBI3* and *IL12A* mainly in B cells, presumably related to Breg cells and Treg (Fig. 7c), as already reported[27–30], demonstrating the presence of IL-35 in tumors. Regarding IL-35R chains, *IL6ST* and *IL27RA* were ubiquitously expressed in immune cells with T cells exhibiting the highest levels for both receptors (Fig. 7d and Supplementary Fig. 7d). In contrast, the expression of *IL12RB2* was confined to lymphocytes and not detected in myeloid cells (Fig. 7d and Supplementary Fig. 7d). The different heterodimers of IL-35R were highly detected in Treg, and a robust co-expression of *IL6ST and IL27RA* and of *IL12RB2* and *IL27RA* was also observed in NK(T) cells (Fig. 7e), conferring NK cells ability to respond to IL-35 within tumors. This observation corroborates our flow cytometry data on blood primary NK cells and validates the relevance of our study in vivo. This is further supported by our observation that the NK(T) cluster displayed the highest enrichment score for our ILC1-like gene signature, as defined by scRNA-seq analysis of IL-35-exposed NK cells (Fig. 7f). In agreement with the expression pattern of *EBI3* and *IL12A* found in public scRNA-seq tumor datasets (Fig. 7b, c), TCGA analysis revealed a positive correlation between *EBI3* and *IL12A* in the majority (20 out of 32) of cancers from the TCGA database (Fig. 7g). These genes were further found to be individually and positively correlated with Treg cells, evaluated through *FOXP3* expression (Supplementary Fig. 7e). Finally, to assess the clinical relevance of our findings, we performed a pan-cancer survival analysis of *EBI3* and *IL12A* expression using the TCGA database including solid tumors only. To avoid confounding effects due to the shared *IL12A* subunit between IL-35 (*IL12A EBI3*) and IL-12 (*IL12A IL12B*) known to be associated to good prognosis, patients were stratified as *IL12A⁺ EBI3⁺ IL12B* (considered as "IL-35" group) *vs* others according to the median of normalized gene

expression. "IL-35⁺" group was significantly associated with poor prognosis for both overall survival (Fig. 7h) and progression-free survival (Fig. 7i) (*p* < 0.0001 each). Altogether, these results support a negative impact of IL-35 in several tumors, that is consistent with its potential role in mediating IL-35R⁺ NK cells loss of antitumor functions and conversion into ILC1-like cells.

## Discussion

IL-35 plays an important role in immune suppression during chronic inflammation by inhibiting effective immune responses while promoting regulatory responses[27,28,30]. However, its effect on NK cell biology remains unknown. In this study, we demonstrated that IL-35 suppresses human primary NK cell effector functions and proliferation while promoting a proangiogenic profile and TGF-β secretion. Furthermore, long-term exposure to IL-35 drives human ConvNK and AdaptNK irreversible conversion into ILC1-like cells with tissue residency features and reduced effector functions through autocrine TGF-β. These findings reveal that IL-35 regulates human NK cell functions and plasticity, extending our knowledge of ILC1 family heterogeneity and providing rationale to target IL-35 to improve NK cell clinical activity.

Our data indicate that short-term exposure of NK cells to IL-35 weakens their activation and production of proinflammatory cytokines and chemokines in response to IL-2, IL-12, IL-18, and IL-15. With IL-35 and IL-12 sharing the IL-12p35 subunit, it has been recently described that IL-35 inhibits IL-12 stimulation by competing with the IL-12Rβ2 receptor[57]. However, this mechanism cannot be applied to IL-2, IL-15, and IL-18, suggesting that IL-35 by itself mediates inhibitory signaling cascades in human NK cells. Furthermore, the downregulated expression of T-BET and EOMES in response to IL-35 could contribute to NK cell inhibition, as they are essential for NK cell effector functions[50,58].

In contrast, IL-35 was able to induce a proangiogenic/TGF-β profile of NK cells, corroborating the reported correlation between IL-35 and increased microvascular density[48,49]. Thus, IL-35 not only inhibits NK cell effector functions but also positively regulates others, as described for other lymphoid populations[28,30]. We also describe the expression of GP130, IL-12Rβ2, and IL-27Rα expression in NK cells that are induced or upregulated upon cytokine activation. Further studies are needed to delineate the composition of the IL-35R in human NK cells. However, as IL-35 was able to inhibit NK cell function in response to IL-15 + IL-18, an activating combination that does not induce GP130 in NK cells, it is tempting to speculate that IL-27Rα and IL-12Rβ2 may be the main components of IL-35R in human NK cells, as described in B cells[59]. Our results shed new light on the immune suppressive activity of IL-35 through NK cell inhibition. This completes the recent observation that selective deletion of IL-35 in Breg cells is associated with an increased NK cell-mediated antitumor immunity in pancreatic cancer models[29].

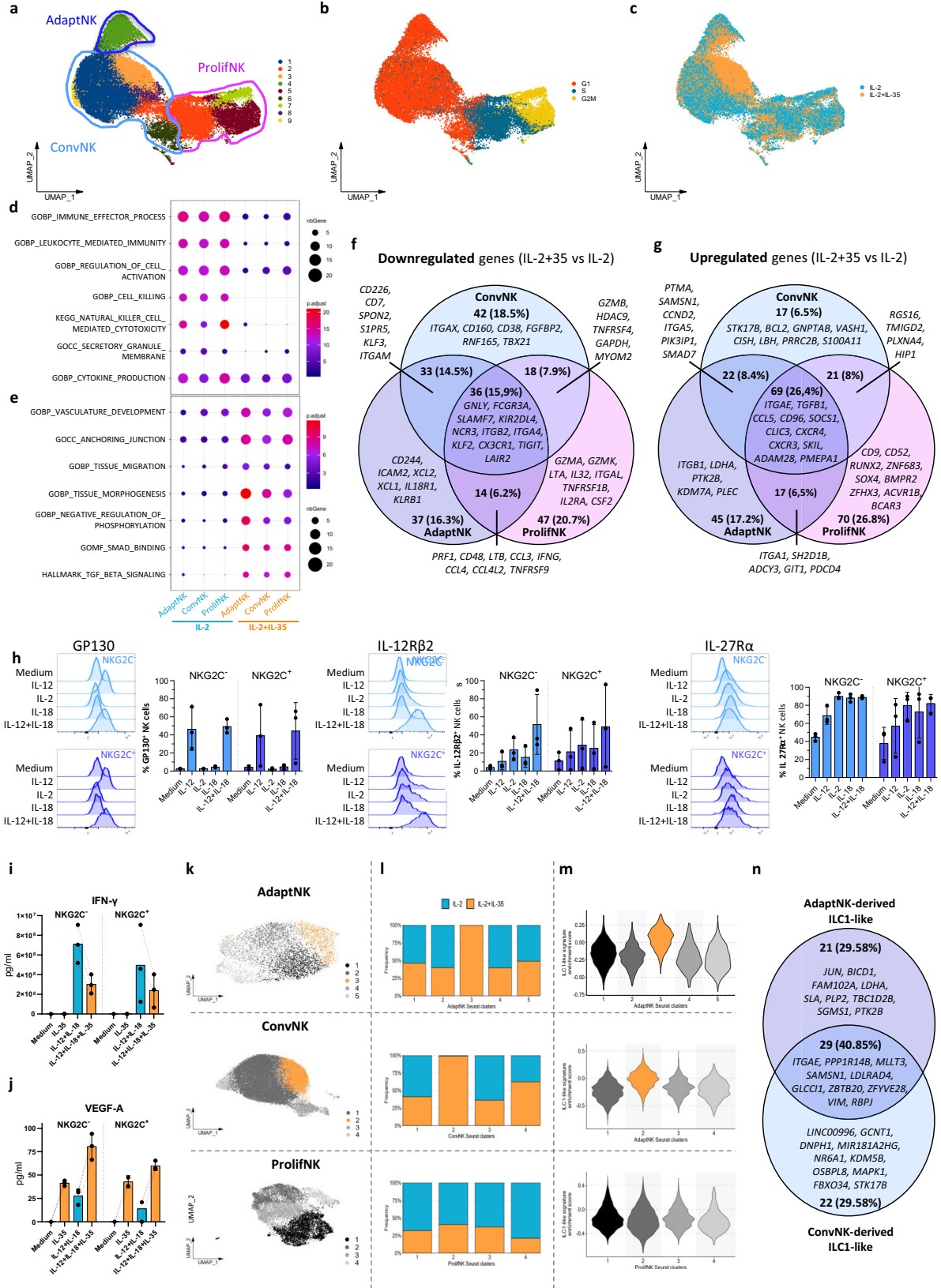

Furthermore, our study revealed that long-term exposure of NK cells to IL-35 inhibits their proliferation and responsiveness to further activation while inducing their co-expression of residency markers (CD9, CD103, and CD49a). This observation is characteristic of the NK-derived ILC1-like phenotype[15,23]. While several studies have associated NK-derived ILC1-like phenotype with decreased proliferative and

cytotoxic potential, either a decreased[23,25] or an increased production[15,16] of IFN-γ, TNF-α or GM-CSF by ILC1-like cells has been reported. These discrepancies may be explained by the nature of the environment, the stimuli used to induce the conversion, the initial NK populations from which the ILC1-like cells derive or an under-appreciated intra-ILC1-like heterogeneity. In this context, our in vitro

**Fig. 4 | scRNA-seq reveals transcriptional regulation and mechanisms involved in NK cell subsets' response to IL-35. a**–**c** UMAP of human purified NK cells cultured in IL-2 with or without IL-35 for 2 or 4 days from two different healthy donors. Cells are colored according to **a** the unsupervised clustering, **b** the proliferation phase, and **c** the treatment received. **d, e** Pathway signatures enriched (Fisher's test adjusted *P*-value < 0.01) in genes up-regulated in absence or presence of IL-35 in ConvNK, AdaptNK and ProlifNK. **f, g** Venn diagram of significantly **f** downregulated or **g** upregulated genes in IL-2 + IL-35-exposed cells compared to IL-2-exposed control cells, by comparing ConvNK, AdaptNK, and ProlifNK being respectively colored in light blue, dark blue and pink. **h** Representative raw histograms of GP130, IL-12Rβ2, IL-27Rα in NKG2C⁺ versus NKG2C⁻ NK cells that were cultured as indicated for 24 h (left). Cumulative histograms of % positive NK cells for each marker (right). Mean values ± S.D are shown (*n* = 3 individual donors).

**i, j** Supernatants from sorted NKG2C⁺ and NKG2C⁻ NK cells cultured for 24 h in the presence of IL-12 and IL-18 with or without IL-35 were collected to quantify **i** IFN-γ and **j** VEGF-A release by ECLIA assay. Results are expressed as pg/ml. Mean values ± S.D. are shown (n = 3 individual donors). **k** UMAP of AdaptNK (top), ConvNK (middle), and ProlifNK (bottom). Cells are colored according to the unsupervised clustering. **l** Frequency of IL-2-treated or IL-2 + IL-35-treated cells in each Seurat cluster among AdaptNK (top), ConvNK (middle) and ProlifNK (bottom). **m** Violin plots showing the distribution across Seurat clusters of the ILC1-like signature score among AdaptNK (top), ConvNK (middle) and ProlifNK (bottom). **n** Venn diagram of the top50 most highly expressed genes among genes specifically expressed in the identified ILC1-like cluster versus other clusters from AdaptNK (dark blue) and ConvNK (light blue). Source data are provided as a Source Data file.

## Table 1 | List of genes used in gene signatures

| Gene signatures | Genes Up | Genes Down |
|---|---|---|
| NK CD56^Bright | *LEF1, TCF7, BACH2, TOX2, GZMK, NCR2, CCR7, CD300C, KIT, IL7R, IL9R, RUNX2, ITGA1, IL1R1* | *ZEB2, ZBTB16, PRDM1, CXCR1, CXCR2, CX3CR1, KIR2DL1, KIR2DL3, KIR3DL1, KIR3DL2, TIGIT, FCGR3A, FCGR3B, S1PR5, CD6, CD160* |
| NK CD56^Dim | *ZEB2, ZBTB16, PRDM1, CXCR1, CXCR2, CX3CR1, KIR2DL1, KIR2DL3, KIR3DL1, KIR3DL2, TIGIT, FCGR3A, FCGR3B, S1PR5, CD6, CD160* | *LEF1, TCF7, BACH2, TOX2, GZMK, NCR2, CCR7, CD300C, KIT, IL7R, IL9R, RUNX2, ITGA1, IL1R1* |
| ILC1-like cells | *CD9, ITGAE, ITGA1, CXCR4, CXCR3, CXCR6, SMAD7, RGS16* | *KLF2, LGALS1, ITGA4, PLEK, S1PR5, CD160* |

conversion experiments demonstrate the coexistence of CD9⁻CD103⁻, CD9⁺CD103⁻, CD9⁻CD103⁺ and CD9⁺CD103⁺ cells illustrating a continuum of conversion. Furthermore, both ConvNK and AdaptNK can undergo ILC1-like conversion upon IL-35 but appear to retain transcriptomic specificities linked to their origin. Taken together, these results suggest the existence of alternative trajectories to become ILC1-like cells as well as an intra-ILC1-like cell heterogeneity.

The capacity of NK cells to adopt an ILC1-like phenotype has been associated with immune dysfunction and disease progression in cancers, infections, and obesity[15,17–19,21,22]. Indeed, since the initial report describing NK cells acquiring ILC1 properties accompanied by a loss of effector and tumor surveillance functions in a mouse model of melanoma[15], accumulating evidence has revealed the presence of cells with phenotypic or functional characteristics of ILC1 within tumors. Recent studies have reported the emergence in cancer patients of both circulating and tumor-infiltrating NK cells expressing at least one of the CD9, CD49a, and CD103 markers. These cells showed impaired effector functions and proliferative ability in favor of the secretion of pro-angiogenic factors[5,15,21,22,60–62]. Similar observations have been made in NK cells cocultured with tumors cells[20,63]. Here, we found that IL-35 is mainly produced in tumor microenvironments by immunosuppressive cells like Treg and a subset of B cells that could be Breg, in agreement with previous works[32,35,48,64]. Thus, IL-35 could locally play a role in the NK conversion into ILC1-like cells in tumor context in vivo. This is particularly relevant as we also provide evidence that IL-35R-expressing NK/ILC1-like cells are present in tumors. As we also report a negative impact of IL-35 in cancer patients' survival, the IL-35-triggered conversion of intratumor NK cells into ILC1-like cells could contribute to tumor progression. Moreover, this IL-35-induced phenotype echoes that of decidual NK cells (dNK)[65–67] which also display reduced cytotoxicity and production of IFN-γ and TNF-α compared to circulating NK cells, and express pro-angiogenic factors such as VEGF-A[65,67–70]. Interestingly, the presence of IL-35-producing decidual Tregs, Bregs and trophoblast suggests a potential involvement of IL-35 in the regulation of NK cells in the decidua[34,42]. Thus, it is tempting to speculate that dNK might actually represent ILC1-like converted NK cells.

The induction of tissue residency features in NK cells by IL-35 mirrors the recently described TGF-β-driven differentiation of CD8⁺ T cells into tissue-resident memory T cells (T_RM), which are specialized in tissue surveillance[71,72]. The attenuation of their effector functions serves as a physiological safeguard to prevent tissue damage. Consistently, autocrine TGF-β has been recently identified as a driver of NK cell residency in mice[73]. Our results suggest that IL-35 might be involved in this process by triggering the production of active TGF-β by NK cells and their consecutive conversion into ILC1-like cells. Interestingly, the overexpression of the *ITGB1* (αvβ1), *ITGA5* (αvβ5), *ADAM28* genes within IL-35-exposed NK cells (Fig. 4g) could constitute putative TGF-β activating proteases[20,74]. We also provide evidence that the IL-35-triggered hyporesponsiveness of NK/ILC1-like cells is not a reversible process, in contrast to the TGF-β-induced one. This suggests that although IL-35 mediated effects are reverted by anti-TGF-β, IL-35 may also have TGF-β-independent activity contributing to the stabilization of their exhausted phenotype.

Regarding the molecular mechanisms, we show that the downregulation of T-BET and EOMES expression, as well as IFN-γ production in NK cells and their conversion into ILC1-like cells, is dependent on autocrine TGF-β production, triggered upon acute or chronic exposure to IL-35. This is consistent with prior studies in NK cells[23,75] and T cells revealing that TGF-β represses T-BET and EOMES, reducing IFN-γ and granzyme levels[23,76,77] and that EOMES must be down-regulated before T_RM express CD103[77].

In summary, we have been able to expand the current knowledge of the immune suppressive role of IL-35 through the negative regulation of NK cell effector functions and their conversion into ILC1-like cells in a TGF-β-dependent manner. We also provide in vivo relevance by demonstrating that IL-35 and IL-35R⁺ NK cells are present in human tumors and that NK(T) cluster displayed enrichment for ILC1-like signature. Nevertheless, additional work in mouse models are needed to assess the role of IL-35 in NK cell regulation during chronic diseases, such as cancers. IL-35 targeting may thus represent a new immunotherapeutic strategy to increase the clinical activity of NK cells.

## Methods
### Human samples
Healthy human blood (collected in EDTA anticoagulant-containing tubes) was purchased anonymously from the French blood agency (Etablissement Français du sang, Lyon France,

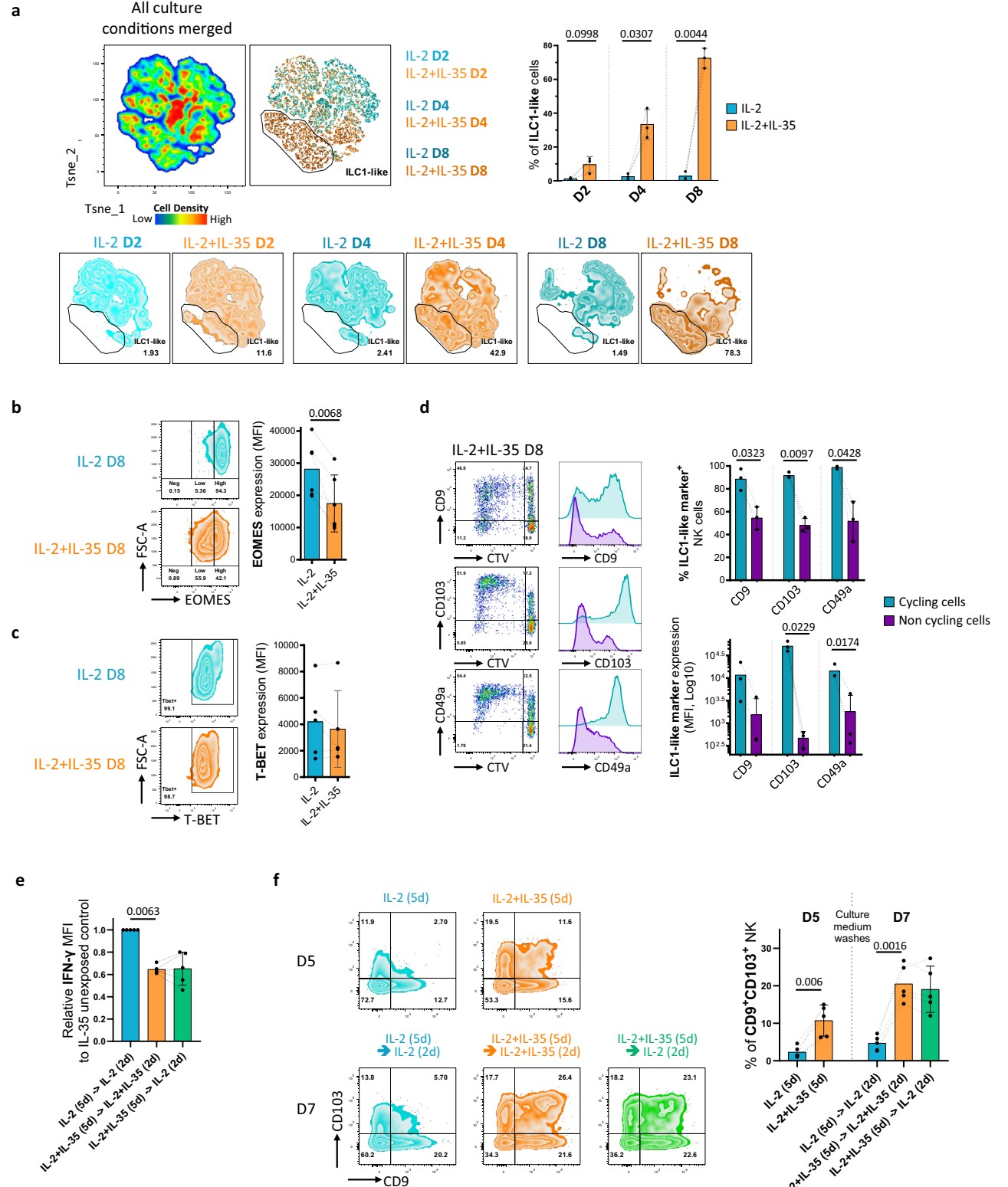

convention 16-093). Participants were French voluntary donors, not recruited specifically for this study and other population characteristics were not collected.

### Human primary NK cells isolation
Peripheral Blood Mononuclear Cells (PBMCs) were isolated from whole blood of healthy donors by density gradient centrifugation on Lymphocyte Separation Medium (Eurobio, ref: CMSMSL01-01). Total

NK cells were purified from PBMCs by negative immune-selection using the Human NK cell isolation kit (Miltenyi, ref: 130-092-657) following the manufacturer's instructions and purity always exceeded 90% (Supplementary Fig. 8a).

### Short-term culture
NK cells were cultured in RPMI GlutaMAX (Gibco, ref: 72400-021) complemented with 10% Fetal Calf Serum, 1% penicillin-

**Fig. 5 | IL-35 drives the conversion of human NK cells into an irreversible ILC1-like phenotype. a** Two-dimensional T-sne plots showing NK cell clustering based on residency surface marker profiles (CD103, CD9, CD49a) after 2, 4, and 8 days in culture (D2, D4, D8, respectively) with IL-2 +/− IL-35. Representative T-sne (upper left and lower) and quantification (%) (upper right) of ILC1-like cells as defined by the combined expression of CD103, CD9 and CD49a. Mean values ± S.D are shown (*n* = 3 individual donors). **b, c** Representative flow cytometry plots (left) and quantification of EOMES and T-BET MFI expression in NK cells cultured for 8 days in the presence of IL-2 with or without IL-35. Mean values ± S.D are shown (*n* = 6 and 5 individual donors, respectively). **d** Representative flow cytometry plots (left) and quantification (right) of % and MFI CD9, CD103, and CD49a expression in ILC1-like

cells after 8 days of culture in the presence of IL-2 with IL-35 according to their proliferation status (cycling vs non-cycling) based on CTV fluorescence intensity. Mean values ± S.D are shown (*n* = 3 individual donors). **e** Quantification of IFN-γ expression in NK cells at day 7 following 48 h (2 d) in indicated culture conditions. Results are expressed as relative MFI compared to control condition IL-2 (5 d) -> IL-2 (2 d). Mean values ± S.D are shown (*n* = 5 individual donors). **f** Representative flow cytometry plots (left) and quantification (%) of CD9⁺ CD103⁺ NK (right) after 5 days (5 d) in culture with IL-2 with or without IL-35 (D5) and at day 7 (D7) following 48 h (2 d) in indicated culture conditions. Mean values ± S.D are shown (*n* = 5 individual donors). Statistical significance was determined using paired T test. Source data are provided as a Source Data file.

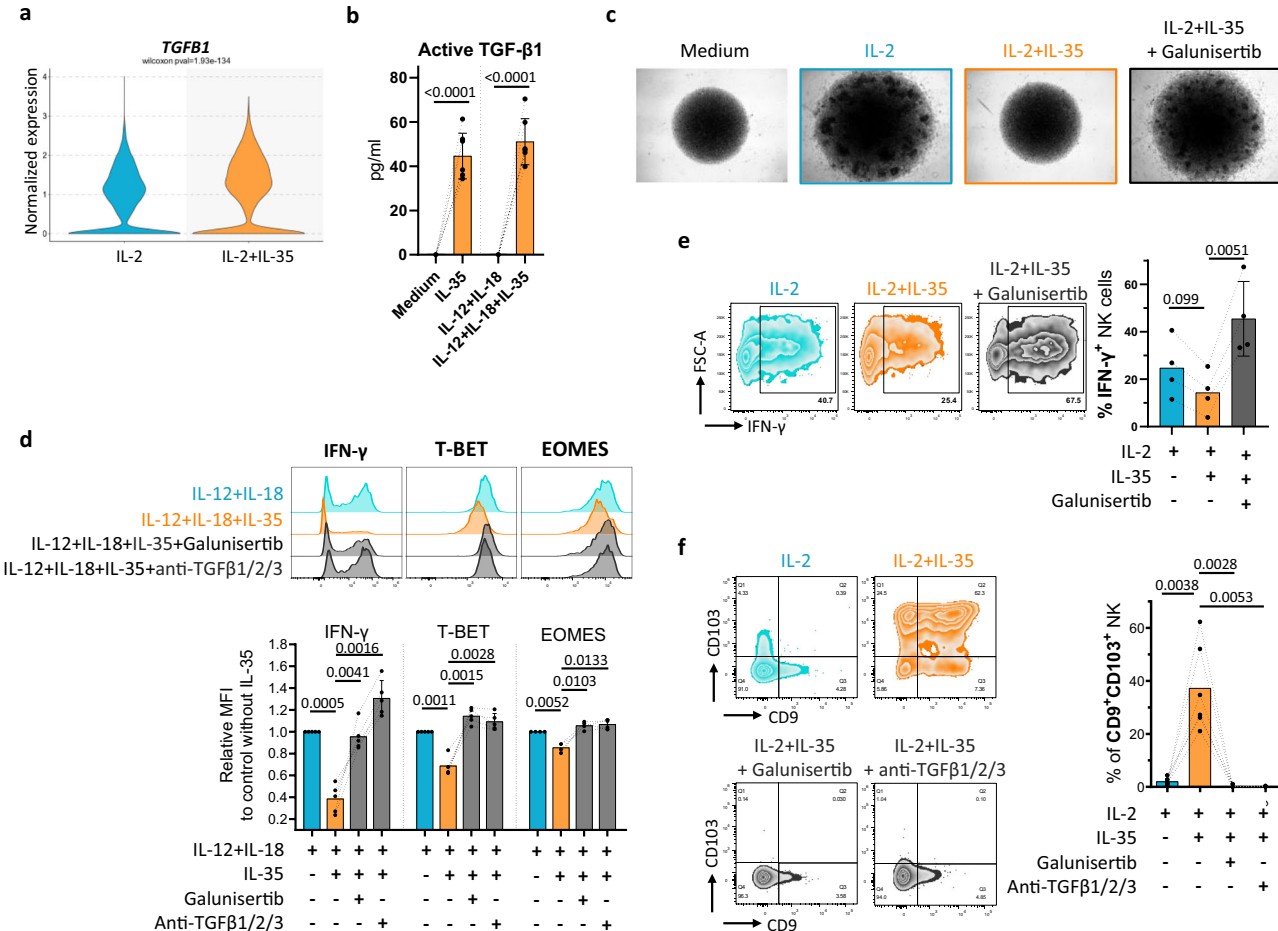

**Fig. 6 | IL-35-triggered autocrine TGF-β drives NK cell dysfunction and conversion into ILC1-like cells. a** *TGFB1* expression in total NK cells cultured for 2 and 4 days in IL-2 with or without IL-35 (analyzed from our scRNA-seq dataset). **b** Supernatants from NK cells cultured for 24 h in the presence of medium versus IL-12 and IL-18 with or without IL-35 were collected to quantify active TGF-β1 by ELISA. Mean values ± S.D are shown (*n* = 7 individual donors). **c** Representative images of NK cells after 7 days of culture in IL-2 with or without IL-35 and a TGF-βR inhibitor (Galunisertib). **d** Representative flow cytometry plots (upper) and quantification of IFN-γ, T-BET, and EOMES expression (lower) in NK cells at 24 h of culture in IL-12 + IL-18 with or without IL-35, TGF-βR inhibitor (Galunisertib) or anti-TGF-β1/2/3 neutralizing antibody. Results are expressed as relative MFI for each marker

compared to control condition without IL-35. Mean values ± S.D are shown (*n* = 4 to 5 individual donors). **e** Representative flow cytometry plots (left) for IFN-γ expression in NK cells after 5 days in culture with IL-2 with or without IL-35 and TGF-βR inhibitor (Galunisertib) and quantification (right) of % IFN-γ⁺ cells in the same culture conditions. Mean values ± S.D are shown (*n* = 4 individual donors). **f** Representative flow cytometry plots (left) for CD9 and CD103 expression in NK cells after 8 days in culture with IL-2 with or without IL-35, TGF-βR inhibitor (Galunisertib) or anti-TGF-β1/2/3 neutralizing antibody and quantification (right) of % CD9⁺ CD103⁺ cells in the same culture conditions. Mean values ± S.D are shown (*n* = 3 to 6 individual donors). Statistical significance was determined using paired T test. Source data are provided as a Source Data file.

streptomycin (Gibco, ref: 15140-122), 1x Non-Essential Amino Acids (Gibco, ref: 11140-035), 1x Sodium Pyruvate (Gibco, ref: 11360-070) (complete RPMI, cRPMI). cRPMI was supplemented with IL-2 (Clinigen, ref: PL1077YFR) (1000 UI/mL), IL-12 (Miltenyi Premium Grade, ref: 130-096-705) (10 ng/mL), IL-15 (Peprotech, ref: 200-15) (10 ng/mL), IL-18 (MBL, ref: B001-5) (10 ng/mL), IL-35 (Peprotech,

ref: 200-37) (100 ng/mL), TGF-β (Biotechne, ref: 240-B) (1 ng/mL) alone or in combination as indicated in figures legends. For blocking experiments, 10 μM Galunisertib (Biotechne, ref: 6956) or control DMSO, 10 μg/mL anti-TGF-β1/2/3 (Biotechne, ref: MAB1835) or mouse IgG1 isotype control (Biotechne, ref: MAB002), was added in the culture medium.

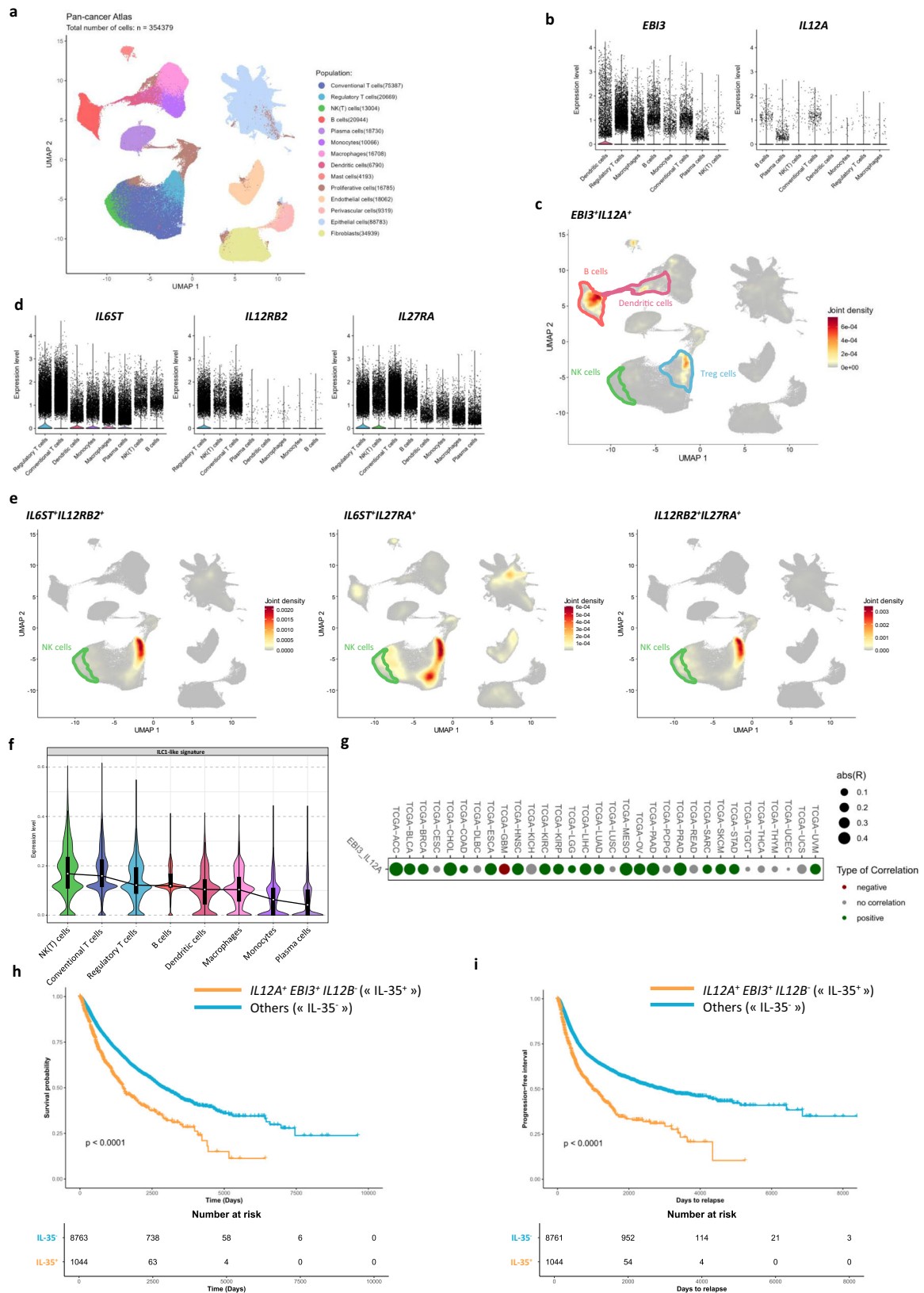

## Human primary NK cells sorting

Purified total NK cells were stained with antibodies for CD56, CD3 and NKG2C, FACS-sorted using a FACSDiscover™ S8 Cell Sorter (BD) (Supplementary Fig. 8b). Cell viability was determined by DAPI staining (1 µg/mL, D1306 Invitrogen). NKG2C+ and NKG2C- NK were collected in separate tubes containing cRPMI and purity always exceeded 98% (Supplementary Fig. 8c, d).

**Fig. 7 | NK/ILC1-like cells expressing IL-35R are present in tumors and IL-35 is associated with poor prognosis in cancer. a** UMAP embedding of a pan-cancer single-cell RNA-seq atlas comprising 354,379 cells, encompassing both immune and non-immune populations following quality control, standard preprocessing, and upstream analysis. **b** Violin plots showing normalized expression levels of IL-35 subunit genes (*EBI3* and *IL12A*) across selected immune cell populations. Expression values are plotted on a shared scale and populations are ranked in descending order based on their average expression level. **c** UMAP embedding of joint expression density for IL-35 cytokine genes, inferred using a Kernel Density Estimator (KDE) to capture co-expression of *EBI3* and *IL12A* at the single-cell level. The joint density corresponds to the weighted average density of individual gene KDEs. **d** Violin plots displaying normalized expression levels of IL-35 receptor components (*IL6ST*, *IL12RB2*, and *IL27RA*) across selected immune cell populations. Expression values are plotted on a shared scale and populations are ranked by mean expression level. **e** UMAP embedding of joint expression density for IL-35 receptor combinations, inferred via Kernel Density Estimation (KDE) to assess co-expression of *IL6ST* with *IL12RB2*, *IL6ST* with *IL27RA*, and *IL12RB2* with *IL27RA*. The joint density corresponds to the weighted average density of individual gene KDEs. **f** Violin plots showing enrichment scores of an ILC1-like transcriptional signature across immune cell populations of interest. Populations are ranked by module enrichment score. **g** Bubble map showing the correlation coefficient for *EBI3* and *IL12A* in TCGA datasets: adrenocortical carcinoma (ACC), bladder urothelial carcinoma (BLCA), breast invasive carcinoma (BRCA), cervical carcinoma (CESC), cholangiosarcoma (CHOL), colorectal adenocarcinoma (COAD), diffuse large B-cell

lymphoma (DLBC), esophageal carcinoma (ESCA), glioblastoma multiforme (GBM), head and neck squamous cell carcinoma (HNSC), kidney chromophobe carcinoma (KICH), kidney clear renal cell carcinoma (KIRC), kidney papillary cell carcinoma (KIRP), lower grade glioma (LGG), liver hepatocellular carcinoma (LIHC), lung adenocarcinoma (LUAD), lung squamous cell carcinoma (LUSC), mesothelioma (MESO), ovarian serous cystadenocarcinoma (OV), pancreatic adenocarcinoma (PAAD), paraganglioma & pheochromocytoma (PCPG), prostate adenocarcinoma (PRAD), rectum adenocarcinoma (READ), sarcoma (SARC), skin cutaneous metastatic melanoma (SKCM), stomach adenocarcinoma (STAD), testicular germ cell cancer (TGCT), thyroid carcinoma (THCA), thymoma (THYM), uterine corpus endometrial carcinoma (UCEC), uterine carcinosarcoma (UCS) and uveal melanoma (UVM). Positive (green) and negative (red) correlation are highlighted, based on p-val<0.05. Dots size represents absR, Pearson correlation coefficient. **h** Patients from TCGA database were stratified as high or low for *EBI3*, *IL12A*, and *IL12B* based on the median and overall survival was analyzed in pan-cancer solid tumors data set. Kaplan-Meier survival curves for patients are represented and p-values were obtained with log-rank test. *EBI3*hi *IL12A*hi *IL12B*lo were classified as IL-35⁺ and compared to other categories. **i** Patients from TCGA database (solid tumors only) were stratified as high or low for *EBI3*, *IL12A*, and *IL12B* based on the median and progression-free survival was analyzed in pan-cancer (solid tumors) data set. Kaplan-Meier survival curves for patients are represented and p-values were obtained with log-rank test. *EBI3*hi *IL12A*hi *IL12B*lo were classified as IL-35⁺ group and compared to other categories.

## Long-term culture and blocking experiment

NK cells were cultured in RPMI GlutaMAX complemented with 20% human AB Serum, 1% penicillin-streptomycin, 1x Non-Essential Amino Acids, 1x Sodium Pyruvate (RPMI + SAB) and supplemented with IL-2 (100 UI/mL), in presence or absence of IL-35 (100 ng/mL). For blocking experiments, 10 μM Galunisertib (Biotechne, ref: 6956) or control DMSO, 10 μg/mL anti-TGFb1/2/3 (Biotechne, ref: MAB1835) or mouse IgG1 isotype control (Biotechne, ref: MAB002), was added in the culture medium. Culture medium was renewed every 2 days for 3 to 8 days. For IL-35 starvation experiments, NK cells were washed 3 times and resuspended in RPMI + SAB supplemented with IL-2 (100UI/mL) for an additional 48 h.

## Proliferation assay

NK cells were stained with 5 μM Cell Trace Violet (CTV) (Invitrogen, ref: C34557) for 20 min at 37 °C, washed twice, activated overnight in RPMI + SAB supplemented with IL-12 (10 ng/mL) and/or IL-15 (10 ng/mL), then washed three times, and cultured for 9 days in RPMI + SAB supplemented with IL-2 (100 UI/mL) in presence or absence of IL-35 (100 ng/mL). Culture medium was renewed every 2 days. CTV dilution was analyzed by flow cytometry on day 3, 5, 7, and 9 and the number of cells at the end of proliferation assay was evaluated using Flow-count Fluorospheres (Beckman Coulter, ref: 7547053).

## Cytotoxic assay

K562 cells were stained with 5 μM CTV following manufacturer' instructions. K562 were co-cultured with NK cells at indicated effector to target ratio for 4 hours at 37 °C. After 4 h, cells were stained with APC Annexin V Apoptosis Detection Kit and PI (Biolegend, ref: 640932), following manufacturer' instructions. NK cytotoxicity was monitored by flow cytometry analysis of K562 cell death.

## Cytokine/Chemokine secretion assay

Cytokines and chemokines were quantified in culture supernatants of activated NK cells. Active TGF-β1 was quantified using LEGEND MAX™ Free Active TGF-β1 ELISA Kit (Biolegend, ref: 437707) following manufacturer's instructions. IFN-γ, TNF-α, GM-CSF, CCL3, CCL4, XCL1, FLT3L, VEGF-A, IL-8, CCL5 were quantified by the MSD technology using the MESO QuickPlex SQ 120 instrument, according to the U-plex protocol, except for XCL1 and CCL5 that were measured

according to the R-plex protocol following manufacturer's instructions.

## Staining and flow cytometry analysis of NK cells

NK cells were incubated with a viability dye (Zombie Nir or Zombie Aqua Biolegend, ref: 423105 or 423101 or Live Dead Blue Invitrogen, ref: L34962) for 20 min at 4 °C. Surface staining was performed using the antibodies listed in Supplementary Table 1 for 15 min at 4 °C. Samples were fixed in 4% formaldehyde solution (Sigma, ref: E7889-100ML) or using the FoxP3/Transcription Factor Staining Buffer Set (eBioscience, ref: 00-5523-00). For IFN-γ intracellular staining, Golgi Plug (BD Biosciences, ref: 555029) was added at 1 μL for 1 million cells in the culture medium, 4 hours before the end of the culture. Following fixation, cells were permeabilized in 0.5% saponin (Sigma, ref: 84510) or using the FoxP3/Transcription Factor Staining Buffer Set (eBioscience, ref: 00-5523-00). Intracellular stainings were performed using the antibodies listed in Supplementary Table 1. After staining, cells were washed twice and samples were acquired on LSR Fortessa flow cytometer (BD Biosciences) or Cytek® Aurora spectral flow cytometer (Cytek). Results were analyzed using FlowJo software (BD v10.8.1).

## scRNA-seq experiment of human NK cells

Purified NK cells from two different healthy donors were cultured in RPMI + SAB supplemented with IL-2 (100 UI/mL) in presence or absence of IL-35 (100 ng/mL). Culture medium was renewed every 2 days. Around 2 million NK cells were collected at days 2 and 4 and washed twice before further processing.

For each collection day, NK cells treated with or without IL-35 were barcoded following Cell Multiplexing Oligos (CMOs) labeling protocol (CellPlex Kit) developed by 10X Genomics. Briefly, NK were independently incubated with specific CMOs, washed 3 times in PBS + 1% BSA and pooled together. The number of live cells was determined with a Luna-FL Dual fluorescence cell counter (Logos Biosystems), loaded on a 10X G chip to obtain an expected cell recovery population of 11,000 cells per channel (≈5000 cells per condition) and run on the Chromium iX system (10X Genomics) according to manufacturer's instructions. Gene Expression and Cell Multiplexing single-cell RNA-seq libraries were generated with the Chromium Single Cell 3' v.3.1 kit with feature barcoding technology (10X Genomics). Gene Expression and Cell Multiplexing libraries were

**Table 2 | Lists of canonical gene symbols used for scoring cell-type-specific gene signatures in public scRNA-seq datasets from the pan-cancer atlas**

| Cell type | Gene Symbol |
|---|---|
| Conventional T cells | *CD3D, CD3E, CD4, CD8A, CD8B, TRDC, SELL, TNF, RORA* |
| Regulatory T cells | *TIGIT, CTLA4, SOD1, TNFRSF4, TNFRSF18, RTKN2, FOXP3* |
| NK cells | *NKG7, NCAM1, XCL1, XCL2, NCR1, GZMA, GZMB, GZMH, PRF1, GNLY, CTSW, KLRB1, KLRD1, KLRF1* |
| ILC1-like cells | *CD9, ITGAE, ITGA1, CXCR4, CXCR3, CXCR6, SMAD7, RGS16* |
| B cells | *MS4A1, CD19, CD79A, CD79B, VPREB3, BANK1* |
| Plasma cells | *IGHG1, IGHA1, MZB1, CD38, JCHAIN* |
| Monocytes | *S100A8, S100A9, FCN1, CSTA, EREG, THBS1, VCAN, LYZ* |
| Macrophages | *C1QA, C1QB, C1QC, GPR34, CSF1R, MS4A4A* |
| Dendritic cells | *CLEC9A, CLEC10A, CLEC4C, XCR1, CD1C, FCER1A, LILRA4, IL3RA, LAMP3, PKIB, INSIG1, IRF4, IRF7, PTCRA, CXCR3* |
| Mast cells | *TPSAB1, KIT, CPA3, SLC18A2, ENPP3* |
| Endothelial cells | *PECAM1, CDH5, KDR, ESAM, EGFL7, VWF, RAMP2, RAMP3, PLVAP, AQP1, EMCN* |
| Perivascular cells | *RGS5, ACTA2, MYH11, MT1M, FRZB, MT1A* |
| Epithelial cells | *EPCAM, KRT5, KRT19, CDH1, SFTPD* |
| Fibroblasts | *COL1A1, COL1A2, MMP2, LUM, PTN, FBLN1, DCN, IGF1, APOD, MEG3, CXCL12* |
| Proliferative cells | *MKI67, STMN1, TOP2A, CDK1* |

sequenced on the NovaSeq 6000 platform (Illumina) to obtain around 50,000 reads and 5000 reads per cell respectively.

## scRNA-seq data QC, normalization, dimensionality reduction, and integration for bioinformatics analysis

Stringent filtering criteria were applied to ensure data quality and informativeness. Genes occurring in fewer than 3 cells, cells with fewer than 500 transcripts, and cells expressing fewer than 500 genes were excluded. Cells harboring more than 10% mitochondrial gene transcripts were removed to mitigate potential effects of low-quality or non-viable cells. To prevent dataset contamination by other cell types, cells with more than 1% Hemoglobin subunit beta and alpha (*HBB/HBA*) genes or a cellular complexity below 80% were also filtered out.

Normalization procedures involved the application of Seurat's NormalizeData function with log-normalization, followed by data scaling using the ScaleData function. Principal Component Analysis (PCA) was subsequently employed for data dimensionality reduction. To address donor-specific effects, data integration was performed using the harmony package, ensuring robust and unbiased downstream analyses.

Upon reduction of dimensions using the UMAP algorithm and the first 20 axes of the PCA, unsupervised clustering with the Leiden algorithm (resolution = 0.5) generated 9 distinct clusters. These clusters were further categorized into three datasets: AdaptNK (cells from cluster 4), ConvNK (cells from clusters 1, 3, 6, 8, and 9), and ProlifNK (cells from clusters 2, 5, and 7). Each dataset underwent renormalization and processing through Harmony. PCA and UMAP (retaining the first 30 axes of the PCA) were used for dimensionality reduction, followed by Leiden clustering at resolutions of 0.6, 0.2, and 0.3, respectively.

## scRNA-seq bioinformatics analysis of NK cells

The initial dataset was processed using CellRanger v7.1.0, facilitating gene expression quantification and sample demultiplexing. Subsequent single-cell analysis was performed utilizing the Seurat R package. Gene signature scores for NK CD56[Bright], CD56[Dim] and ILC1-like cells were computed using the UCell R package. Two separate scores were calculated: one using the list of upregulated genes and another using the list of downregulated genes, for each individual cell (see lists in Table 1). To obtain a composite signature score, the UCell enrichment score of the downregulated gene set was subtracted from that of the upregulated gene set. Pathway enrichment analysis of cells was conducted using the UCell R package. Marker genes for all clusters

were identified using the FindAllMarkers function from Seurat, considering only positively expressed genes with a minimum percentage of expressing cells of 0.5. Differential analyses between IL-2 and IL-2 + IL-35 were performed using the FindMarkers function from Seurat and the MAST algorithm. Genes with a *p*-value < = 0.5 were selected, and an over-enrichment analysis of up- or downregulated genes was conducted using the enricher function of the clusterProfiler R package and GO and KEGG pathways from MSigDB.

## Analysis of public cancer patient scRNA-seq data

GSE131907 (lung[53]), GSE160269 (esophagus[54]), GSE166555 (colon[55]) and GSE176078 (breast[56]) public scRNA-seq datasets were obtained from the original publications. The same quality controls metrics thresholds as described in the original publications was applied and only the cells from primary tumor of patients without treatment was retained for each study.

For each dataset, the standard pipeline Seurat (v5.1.0)[78] was perform for data normalization, dimensionality reduction and clustering using parameters of the original publications. The summary of each parameter for quality control, standard preprocessing and downstream analysis is resumed in Supplementary Table 2.

Shared Nearest Neighbors (SNN) graph, Louvain clusters and UMAP embeddings were calculated for each study and adapted from the author annotation provided in the metadata. RunHarmony function[79] was applied to remove the intra-study batch effect. After validation of the clustering step, the different cell types were re-annotated on the basis of canonical gene score signatures using the UCell package[80] in order to harmonize cell populations between the different datasets (see Table 2).

Next, the 4 datasets were merged without applying an additional filtering step, resulting in a total of 354379 cells of 4 different anatomical sites (lung[53] GSE131907; Esophagus[54] GSE160269; Colon[55] GSE166555; Breast[56] GSE176078) from 108 patient sample. The merged data were reprocessed using the standard Seurat pipeline according to the layer-based Seurat v5.

Dimensional reduction was performed using the RunUMAP function and the first 30 principal components (PC) and the neighbor graph was constructed using the FindNeighbors function, on the first 30 PC. The FindClusters function was applied to identify clusters with a resolution set to 0.6, resulting in consistent clusters according to the harmonized annotation of the different cell types.

Normalized expression levels were represented as violin plot with VlnPlot function and bubble plot with DotPlot function. Density plot

was generated using the scCustomize package[81] and the Plot_Density_Joint_Only function that enriches the analysis of single-cell data with custom visualization functions and acts here as a wrapper around the plot_density function of the Nebulosa package[82] to visualize gene expression levels as densities on dimension reduction projections such as UMAP.

### Analysis of public cancer patient RNA-seq data (TCGA)

Upper-quartile normalized expression (UQN) and clinical outcome datasets from The Cancer Genome Atlas (TCGA) were downloaded from the Pan-Cancer Atlas (https://www.cancer.gov/about-nci/organization/ccg/research/structural-genomics/tcga) for 32 tumor types: adrenocortical carcinoma (ACC), bladder urothelial carcinoma (BLCA), breast invasive carcinoma (BRCA), cervical carcinoma (CESC), cholangiosarcoma (CHOL), colorectal adenocarcinoma (COAD), diffuse large B-cell lymphoma (DLBC), esophageal carcinoma (ESCA), glioblastoma multiforme (GBM), head and neck squamous cell carcinoma (HNSC), kidney chromophobe carcinoma (KICH), kidney clear renal cell carcinoma (KIRC), kidney papillary cell carcinoma (KIRP), lower grade glioma (LGG), liver hepatocellular carcinoma (LIHC), lung adenocarcinoma (LUAD), lung squamous cell carcinoma (LUSC), mesothelioma (MESO), ovarian serous cystadenocarcinoma (OV), pancreatic adenocarcinoma (PAAD), paraganglioma & pheochromocytoma (PCPG), prostate adenocarcinoma (PRAD), rectum adenocarcinoma (READ), sarcoma (SARC), skin cutaneous metastatic melanoma (SKCM), stomach adenocarcinoma (STAD), testicular germ cell cancer (TGCT), thyroid carcinoma (THCA), thymoma (THYM), uterine corpus endometrial carcinoma (UCEC), uterine carcinosarcoma (UCS) and uveal melanoma (UVM) to analyze *EBI3, IL12A, IL12B, FOXP3* gene expression in tumors. Patients with solid tumors were stratified according to the median of normalized gene expression to obtain "high" and "low" groups. *IL35*+ group were then determined as *EBI3_High + IL12A_High + IL12B_Low*, and other combination as *"Other"*. For progression-free and overall survival analyses were performed with R software v 4.4.2, using the packages survival v3.7-0 and survminer v0.5.0. The log-rank test was used to determine statistical significance between groups of patients. Pearson's correlation between *EBI3, IL12A and/or IL12B* expression levels was computed and visualized using scatter plots with linear regression lines generated via the ggplot2 v3.5.1 package.

### Quantification and statistical analysis

Statistical analyses were performed using the GraphPad software and tests conducted as well as exact p-values are indicated in the figure or figure legends legends. P-values lower than 0.05 were considered to be significant.

### Reporting summary

Further information on research design is available in the Nature Portfolio Reporting Summary linked to this article.

## Data availability

The scRNA-seq data generated during this study are available in Gene Expression Omnibus repository (GEO) database under accession code GSE256137. The public scRNA-seq data used in this study are available in GEO database under accession code GSE131907, GSE160269, GSE166555, and GSE176078. Source data are provided with this paper.

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

## Acknowledgements

Single cell-RNA sequencing was performed by the genomic platform of the CRCL supported by the SiRIC-LYriCAN program (grant INCa-DGOS-INSERM-ITMO cancer_18003). Cell sorting was performed at the flow cytometry facility of the CRCL. We thank the members of the flow cytometry platform, T. Andrieu, P. Battiston-Montagne and A. Jambon. We would like to express our sincere gratitude to the T-LYACTS team (CIRI INSERM U1111 Lyon France) for so kindly welcoming us into their laboratory when we faced challenges in our own. This work was supported by the Fondation ARC pour la Recherche sur le Cancer (grant PJA20181208305 to N.B.-V.), the Institut National Du Cancer (INCA; AAP PLBIO-16-116 to N.B.-V.), the Ligue Régionale contre le cancer (Comité du Rhône) (to N.B.-V.), the Ligue Nationale contre le cancer (EL2020.LNCC-CHC to C.C.), the LABEX DEVweCAN (ANR10-LABX-0061 to C.C.) of Université de Lyon, within the program "Investissements d'Avenir" (ANR-11-IDEX-0007) operated by the French National Research Agency (ANR) to C.C.). V.P. is a recipient of a doctoral fellowship from the French Government PhD Fellowship (2019-2022) and 1-year extension Ph.D fellowship from the Fondation ARC. L.R-B. is a recipient of a doctoral fellowship from the French Government PhD Fellowship (2022-2025). T.C. is a recipient of a doctoral fellowship from MSD Avenir.

## Author contributions

V.P., C.D., C.C., and N.B.-V. conceived and designed the study; V.P. and C.D. generated the scRNA-seq data; L.T., A.F., and V.P. analyzed the scRNA-seq data; T.C. and A.V. performed the bioinformatics analysis of public datasets; V.P., L.R.-B., C.R., D.P., E.S., and U.H., performed and analyzed the in vitro experiments; C.C. and N.B.-V. supervised the research; V.P., L.R.-B., L.T., E.P., C.D, C.C., and N.B.-V. wrote the manuscript; and all authors discussed the data and the manuscript.

## Competing interests

The authors declare no competing interests.
