## [Transparent Peer Review file · Nature Communications]

Interleukin-35 impairs human NK cell effector functions and induces their ILC1-like conversion with tissue residency features

Corresponding Author: Dr Nathalie Bendriss-Vermare

Version 0:

Reviewer comments:

Reviewer #1

(Remarks to the Author)

In the present study, Picant et al. sought to determine the role of IL-35 in human NK cells by supplementation in vitro with IL-35 combined with other activating cytokines. Short-term IL-35 stimulation decreased NK cell activation, accompanied by decreased effector cytokine production and increased TGF- β production. Prolonged stimulation with IL-35 inhibits NK cell proliferation and cytotoxic functions. We next investigated whether IL-35 is involved in the TGF- β -mediated conversion of NK cells into ILC1-like cells and showed that longer exposure to IL-35 can induce an ILC1-like phenotype in a TGF- β -dependent manner. Finally, single cell RNA sequencing further characterized the gene expression of the IL-35-induced cell population with an ILC1-like gene signature.

The role of IL-35 in chronic human diseases in vivo remains unclear in the present study; however, considering previous reports, an immunoregulatory role of IL-35 has been suggested. The reviewer is concerned about the lack of mechanistic investigations into how IL-35 acts on NK cells, resulting in short or long-term changes in their functions. Therefore, further experiments and significant revisions should be carried out to strengthen the conclusions of this study. Specific comments are as follows:

Major Comments:

1. The abstract does not adequately justify the focus on NK cells, explain the importance of the regulatory mechanisms uncovered, or delineate between novel findings and existing knowledge. In conclusion, the authors overstate that "our findings identify a new role for IL-35 as a key driver of NK cell plasticity. However, this claim is not supported because only in vitro experiments were conducted.
2. It remains unclear whether IL-35 acts directly on IL-35 receptors on the surface of NK cells. The authors evaluated the expression of IL-35 receptors in NK cell subpopulations. Employing genetic modifications (e.g., siRNA or CRISPR-Cas9-mediated knockout) targeting IL-35 receptors in NK cells would further support the conclusions of the authors.
3. In Fig. 4e and Supplementary Figs. 4e and 4f, the authors concluded that autocrine TGF- β induced CD9+CD103+ NK cells using the TGF β 1 signaling inhibitor, galunisertib. This result raises concerns about the specificity of the inhibitor. The authors should perform experiments using more specific approaches such as neutralizing antibodies.
4. The authors first showed that short-term (16 h) IL-35 stimulation decreased the expression of the effector cytokines T-bet and Eomes (Fig. 1). However, the molecular mechanisms underlying these acute changes remain unclear. Furthermore, the effect of acute exposure on cell viability under each condition should be evaluated.
5. Given the heterodimeric structure of IL-35, the possibility that the recombinant human IL-35 protein used in the experiments may contain nonheterodimerized p35 and IL-27 subunits should be considered. The authors should evaluate whether IL-35 from alternative sources (e.g., other products) shows the same tendency.
6. Regarding the scRNA-seq dataset shown in Fig. 5, the authors should disclose the number of healthy donors included in the dataset.
7. The authors should explain how they calculated the CD56Dim NK, CD56Bright NK, and ILC1-like gene signatures in the Methods section.

Minor comments:

1. The manuscript also lacks linguistic deficiencies, but more importantly, the logical flows across the paragraphs obscure the intended message. Additionally, it is unclear whether the information presented in the literature pertains to humans or

animals, necessitating clarification throughout the introduction section.

2. As shown in Fig.1a, quadrant gating of the FACS plots differed across NK cells treated with or without IL-35, especially in the treatment conditions of IL-12 + IL-2, IL-12 + IL-18, and IL-15 + IL-18.

3. In Fig. 1, the authors stimulated NK cells with 1000U / ml IL-2, but with 100U/ml in Figs. 2 to 5. The authors explain why they used different conditions.

4. In all the pseudo-colored t-SNE plots, the pseudo-color should be described.

5. In the sentence explaining Fig. 5d: 'In accordance with our previous data, gene set enrichment analyzes...' The reference seemed missing.

6. In the last sentence of page 7, "with 55.5 % and 50.5% of total genes being, respectively, downregulated or upregulated by IL-35 in only one NK population". It is not clear what data this sentence indicates.

Reviewer #2

(Remarks to the Author)

The manuscript by Picant et al. elucidates the role of IL-35, a member of the IL-12 family, on human NK cells. The authors show that exposure of human NK cells to IL35 in combination with IL12 and IL18 results in decreased frequencies of IFN-g expressing cells, decreased activation markers, decreased TNF, GM-CSF and chemokine production, but to enhanced VEGF production and TGF- β . This phenotype correlates with decreased expression of Eomes. Moreover, IL-35 inhibits cytokine induced NK cell proliferation, but not survival, and extended exposure results in low NK cell anti-tumor reactivity correlating with development of an ILC1 phenotype. The direct impact of IL-35 on human NK cells and its function dampening action is a novel aspect for NK cell biology, an important finding and of interest to the field. The methodology is sound. However, all experiments are performed in vitro by addition of IL-35 to other cytokine combinations and the physiological role of these findings needs to be better elucidated.

- Fig 1: Since both adaptive and conventional NK cells react to IL-35, can the authors stain for the IL-35 receptor expressed on NK cells? Do both subsets produce TGF- β ? and VEGF? How is the IL-35 receptor regulated? How is the signaling via the receptor impacting the dampening immune responses? Does IL12 partially compete? How does IL-35 cause the delay in NK cell proliferation? Additional data should support these findings.

- Fig 3: The authors show hyporesponsiveness of NK cells exposed long term to IL-35. Is this hyporesponsiveness reversible? Is NK cell produced TGF β responsible for this phenotype?

- Since the authors claim an importance of their findings in the tumor microenvironment, it would be highly relevant to investigate the presence of IL-35 (its sources) and NK cells/ILC1 signatures in human tumors and to correlate these to tumor progression and patient survival.

- How do the amounts of IL-35 relate to the amounts produced in vivo?

Minor:

- The experiments are performed with distinct cytokine combos. What happens to NK cell mediated tumor killing if NK cells are exposed to tumor cells in the context of IL-35?

- The authors mention that their NK cell preparations are 95%. They should stain for the remaining cell populations and show these data.

Version 1:

Reviewer comments:

Reviewer #1

(Remarks to the Author)

The authors have made significant efforts to address the reviewer's concerns and improve the manuscript. In particular, the additional analyses of IL-35-producing cells and IL-35 receptor-expressing NK cells in human tumors, along with the correlation with IL-35 levels and clinical prognosis, strengthen the significance of the study. The revised manuscript provides new insights into IL-35-mediated human NK cell regulation in tumor microenvironment and is worth publishing.

Reviewer #2

(Remarks to the Author)

My comments were addressed to my satisfaction and the data reanalysing scRNA seq of cancer patients enriched the manuscript. I noted that the legend of Fig. 7f was too small to read.

Reviewer #1 (Remarks to the Author):

In the present study, Picant et al. sought to determine the role of IL-35 in human NK cells by supplementation *in vitro* with IL-35 combined with other activating cytokines. Short-term IL-35 stimulation decreased NK cell activation, accompanied by decreased effector cytokine production and increased TGF- β production. Prolonged stimulation with IL-35 inhibits NK cell proliferation and cytotoxic functions. We next investigated whether IL-35 is involved in the TGF- β -mediated conversion of NK cells into ILC1-like cells and showed that longer exposure to IL-35 can induce an ILC1-like phenotype in a TGF- β -dependent manner. Finally, single cell RNA sequencing further characterized the gene expression of the IL-35-induced cell population with an ILC1-like gene signature. The role of IL-35 in chronic human diseases *in vivo* remains unclear in the present study; however, considering previous reports, an immunoregulatory role of IL-35 has been suggested. The reviewer is concerned about the lack of mechanistic investigations into how IL-35 acts on NK cells, resulting in short or long-term changes in their functions. Therefore, further experiments and significant revisions should be carried out to strengthen the conclusions of this study. Specific comments are as follows:

We really thank the reviewer for this thorough review and valuable and constructive feedback on our study. We appreciate the opportunity to address these concerns and provide additional context and further clarification.

We understand the point regarding the need for further mechanistic investigations into how IL-35 acts on NK cells. While our current study primarily focuses on the phenotypic and functional changes induced by IL-35, our main objective was first to investigate the effects of IL-35 on human NK cell biology, that was never addressed in the literature, and how it participates in NK dysfunction that was also poorly understood. That said, we agree that elucidating the underlying mechanisms is crucial for a comprehensive understanding. To address this, we conducted additional experiments to investigate the signaling pathways and molecular interactions involved in IL-35-mediated regulation of NK cells. Specifically, we have:

- explored more in depth our scRNAseq data to identify biological pathways regulated by IL-35 in NK cells that led to a reorganization of our manuscript, as scRNAseq being the starting point for biological (ILC differentiation, NK subsets, etc.) and mechanistic (receptor expression, role of TGF- β and its activation, etc.) explorations.
- investigated the expression of the 3 chains of different IL-35R and their regulation upon NK cell activation *in vitro* by flow cytometry and completed/confirmed the results with *in silico* analyses of human public data.
- examined the intracellular signaling pathways activated by IL-35 through phospho-flow cytometry.
- performed knockdown studies by using CRISPR-Cas9 experiments to target the 3 chains of IL-35R and to determine their respective role in the observed phenotypic and functional changes.
- investigated the impact of IL-35 on the expression of T-bet and Eomes transcription factors known to be involved in NK cell differentiation and function in response to acute exposure to IL-35 (24h)
- demonstrated in more details the effects of IL-35-induced TGF- β and the role of autocrine TGF- β in the downregulation of Eomes and T-bet expression and IFN- γ production in NK cells in short-term (acute exposure) as well as long-term (chronic exposure)) cultures by using TGF- β blocking molecules (galunisertib and anti-TGF β 1/2/3 antibodies).

-discussed more in depth the potential underlying molecular mechanisms based on existing knowledge and literature.

Finally, while our current study is limited to *in vitro* experiments, we recognize the importance of validating our findings *in vivo* to assess the role of IL-35 in NK cell regulation during chronic diseases. To provide new insights about the *in vivo* relevance of our *in vitro* findings, we used publicly available single-cell RNAseq datasets of human tumors (see also response to point 1). By merging these datasets, we identify the main sources of IL-35 in human tumors and demonstrate that IL-35R+ NK cells are present in tumors and enriched in an ILC1-like gene signature. We also demonstrate a negative impact of IL-35 on clinical outcome in a TCGA pancancer analysis. These new results added as new Figure 7 provide new insights about the *in vivo* relevance of our *in vitro* findings.

We have incorporated these additional experiments and revised our manuscript to include the new data and insights gained from these mechanistic investigations. We believe these revisions will significantly strengthen our conclusions and address the reviewers' concerns.

Major Comments:

1. The abstract does not adequately justify the focus on NK cells, explain the importance of the regulatory mechanisms uncovered, or delineate between novel findings and existing knowledge. In conclusion, the authors overstate that "our findings identify a new role for IL-35 as a key driver of NK cell plasticity. However, this claim is not supported because only *in vitro* experiments were conducted.

We agree with the reviewer and accordingly we have revised the abstract to highlight the significance of NK cells in the context of our study, clearly delineate between novel findings and existing knowledge, emphasize the regulatory mechanisms uncovered and their implications.

We understand the reviewers' concern regarding the strength of our claim about IL-35's role in NK cell plasticity. While our *in vitro* experiments provide valuable insights, we agree that further *in vivo* studies are necessary to fully support this claim. We have revised the conclusion to more accurately reflect the scope of our findings and to acknowledge the need for additional research. Specifically, we have stated that our *in vitro* findings suggest a potential new role for IL-35 in NK cell plasticity, and that future *in vivo* studies are warranted to confirm these observations. Nevertheless our data fit into a very logical continuity of recent, high-impact, and very important papers in recent years in understanding NK biology.

Furthermore, by integrating four public single-cell RNAseq datasets into a harmonized cell atlas covering >350k cells across 108 patients' samples, we analyzed the expression of IL-35 genes (*EBI3* and *IL12A*) and IL-35R chains (*IL6ST*, *IL12RB2*, *IL27RA*) in human tumors. These new results identify Breg, Treg, and to a lesser extent Dendritic cells as the main source of IL-35 in human tumors and demonstrate that NK cells expressing *IL12RB2* and *IL27RA*, and to a lesser extent *IL6ST*, are present in tumors and enriched in an ILC1-like gene signature. We also demonstrate a negative impact of IL-35 on clinical outcome in a TCGA pancancer analysis. These new results added as new Figure 7 provide new insights about the *in vivo* relevance of our *in vitro* findings.

2. It remains unclear whether IL-35 acts directly on IL-35 receptors on the surface of NK cells. The authors evaluated the expression of IL-35 receptors in NK cell subpopulations. Employing genetic modifications (e.g., siRNA or CRISPR-Cas9-mediated knockout) targeting IL-35 receptors in NK cells would further support the conclusions of the authors.

We developed a spectral flow cytometry panel to evaluate the expression of the 3 known chains of IL-35R (gp130, IL12Rb2, and IL27Ra) in NK cells, at basal levels and upon activation by different cytokines alone (IL-12, IL-15, IL-18) or in combination 2 by 2. The gp130 chain was not expressed in resting NK cells but its expression was specifically induced by IL-12 in 40% of NK cells and the addition of IL-15 or IL-18 does not change the expression levels (new Fig. 1g). Very few NK cells (5-10%) constitutively expressed IL12Rb2, while IL27Ra was expressed in 35-45% of total NK cells. IL12Rb2 was upregulated in response to individual cytokines and to a greater extent when cytokines were combined. IL27Ra is strongly upregulated in NK cells in response to IL-2, IL-15 and IL-18 and to a lesser extent to IL-12 to reach >90% IL27Ra positive NK cells in response to cytokine combination (new Fig. 1g).

The expression pattern and regulation of gp130, IL12Rb2, and IL27Ra was similar between NKG2C-conventional and NKG2C+ adaptive NK cells. Furthermore, the three receptors are regulated in a similar manner upon NK cell subsets activation by cytokines used alone (IL-12, IL-15, IL-18) or in combination 2 by 2 (new Fig. 4h).

We have also used in-house transcriptomic datasets of primary NK cells that confirmed the flow cytometry data (data not included in the manuscript).

As there is no validated tool to neutralize the activity of human IL-35, a heterodimeric protein composed of IL-12 α (P35) and IL-27 β chains, we performed CRISPR-Cas9 mediated KO of IL-35R chains (gp130, IL12Rb2, IL27Ra) in human primary NK cells. We selected 3 different gRNA per receptor (pre-designed by IDT and pre-validated in the NK92 cell line) and electroporated NK cells using the Neon electroporation device. Despite good viability of electroporated NK cells after 48h culture in low dose IL-2, genome editing was not efficient enough to generate IL-35RKO NK cells to investigate the role of the 3 chains in NK cell response to IL-35.

3. In Fig. 4e and Supplementary Figs. 4e and 4f, the authors concluded that autocrine TGF- β induced CD9+CD103+ NK cells using the TGF β 1 signaling inhibitor, galunisertib. This result raises concerns about the specificity of the inhibitor. The authors should perform experiments using more specific approaches such as neutralizing antibodies.

As requested by the reviewer, we repeated the experiments using specific anti-TGF β 1/2/3 neutralizing antibodies (R&D MAB1835). Anti-TGF β 1/2/3 neutralizing antibody reverted the inhibitory effect of IL-35 as galunisertib did (new supp Fig 6e and new Fig. 6d and f), confirming the specificity of the inhibitor. Indeed, using anti-TGF- β antibody or galunisertib, we showed that blocking autocrine TGF- β 1 prevented the acquisition of tissue residency markers in presence of IL-35 (Fig. 6f and Supplementary Fig. 6e) and the downregulation of Eomes and IFN- γ expression in long-term cultures (Supplementary Fig. 6f), as previously observed with galunisertib. We also now show that anti-TGF- β 1/2/3 antibody prevents the downregulation of Eomes, Tbet, and the inhibition IFN- γ production in short term culture with IL-35 (new Fig 6d).

4. The authors first showed that short-term (16 h) IL-35 stimulation decreased the expression of the effector cytokines Tbet and Eomes (Fig. 1). However, the molecular mechanisms underlying these acute changes remain unclear.

Regarding the molecular mechanisms, we now show that the downregulation of Tbet and Eomes expression, as well as IFN- γ production, is dependent on autocrine TGF- β production by NK cells that is triggered upon acute or chronic exposure to IL-35 (added as new figure 6). Our observation is consistent with prior studies in NK cells (Cortez et al, Immunity 2016 ; Harmon et al, Front Immunol

2019) and T cells revealing that TGF- β represses Eomes and T-bet, reducing IFN- γ and granzyme levels (Thomas and Massague, 2005 ; Mackay et al., 2015; Cortez et al., 2016) and that EOMES must be down-regulated before resident memory T cells express CD103 (Mackay et al., 2015). Furthermore, ChIP-Seq data from *in vitro* activated CD8 T cells (GSE135533) revealed that the promoter region for the Eomes gene includes multiple SMAD4 binding sites (Chandiran et al, 2022). However, it was shown that Eomes expression was suppressed in T cells by TGF- β via the c-Jun N-terminal kinase (JNK)-c-Jun signaling pathway (Ichiyama et al, 2011). These articles have been added in the discussion to provide clues for potential molecular mechanisms underlying the acute changes triggered by IL-35 in NK cells.

Furthermore, the effect of acute exposure on cell viability under each condition should be evaluated.

In the first version of our manuscript we reported that chronic exposure to IL-35 did not alter NK cell viability at different time points (day3, 5, 7, and 9) during long-term cultures (see figure 2e). New supp Fig 1c now shows that there is no effect either of acute exposure on NK cell viability under each condition.

5. Given the heterodimeric structure of IL-35, the possibility that the recombinant human IL-35 protein used in the experiments may contain nonheterodimerized p35 and IL-27 subunits should be considered. The authors should evaluate whether IL-35 from alternative sources (e.g., other products) shows the same tendency.

We have used two different commercially available sources of recombinant human IL-35 (Miltenyi and Peptotech) that provided similar results.

Furthermore, by using another source of recombinant human IL-35 from Chimerin Laboratories, Li S *et al* found that IL-35 treatment reduced NK cytotoxicity (NK92 cell line was used, not primary NK cells) against K562, a typical NK target cell, in total agreement with our observations (Nature 2022 PMID: 36198789).

6. Regarding the scRNA-seq dataset shown in Fig. 5, the authors should disclose the number of healthy donors included in the dataset.

The number of donors included in the scRNA-seq dataset has been added in new Fig. 4 (n=2).

7. The authors should explain how they calculated the CD56Dim NK, CD56Bright NK, and ILC1-like gene signatures in the Methods section.

We apologize for omitting this information. Additional information about the signatures are now provided in the Methods section.

Minor comments:

1. The manuscript also lacks linguistic deficiencies, but more importantly, the logical flows across the paragraphs obscure the intended message. Additionally, it is unclear whether the information presented in the literature pertains to humans or animals, necessitating clarification throughout the introduction section.

We appreciated the reviewers' insights and have addressed the points raised to improve the manuscript. Firstly, we have reviewed the manuscript for any linguistic deficiencies and ensured that

the language is clear and precise. Secondly, we have worked on enhancing the logical flow across the paragraphs to make the intended message more apparent. This has involved restructuring some figures and sections and ensuring that each paragraph transitions smoothly to the next. Lastly, we have clarified whether the information presented in the literature pertains to humans or animals. We have made sure to specify this throughout the introduction and discussion sections to avoid any confusion.

2. As shown in Fig.1a, quadrant gating of the FACS plots differed across NK cells treated with or without IL-35, especially in the treatment conditions of IL-12 + IL-2, IL-12 + IL-18, and IL-15 + IL-18.

As the IFN- γ negative population shifts according to activating conditions, we always gated according to the specific control of the condition (with vs without IL-35).

3. In Fig. 1, the authors stimulated NK cells with 1000U / ml IL-2, but with 100U/ml in Figs. 2 to 5. The authors explain why they used different conditions.

We used different concentrations of IL-2 depending on the biological readouts, as commonly used in the NK field. Indeed, 1000 UI/ml IL-2 was used to activate NK cells in short term cultures while we used 100 UI/ml IL-2 for NK cell proliferation assay to avoid activation-induced cell death in long term experiments.

4. In all the pseudo-colored t-SNE plots, the pseudo-color should be described.

We apologize for omitting this information. The pseudo-color bar has been added in each pseudo-color t-SNE plots.

5. In the sentence explaining Fig. 5d: 'In accordance with our previous data, gene set enrichment analyzes...' The reference seemed missing.

This sentence was referring to our data included in this paper. This has been clarified by quoting the numbers of the figures (Fig 2,3,4).

6. In the last sentence of page 7, "with 55.5 % and 50.5% of total genes being, respectively, downregulated or upregulated by IL-35 in only one NK population". It is not clear what data this sentence indicates.

This has been clarified in the text.

Reviewer #2 (Remarks to the Author):

The manuscript by Picant et al. elucidates the role of IL-35, a member of the IL-12 family, on human NK cells. The authors show that exposure of human NK cells to IL-35 in combination with IL12 and IL18 results in decreased frequencies of IFN-g expressing cells, decreased activation markers, decreased TNF, GM-CSF and chemokine production, but to enhanced VEGF production and TGF- β . This phenotype correlates with decreased expression of Eomes. Moreover, IL-35 inhibits cytokine induced NK cell proliferation, but not survival, and extended exposure results in low NK cell anti-tumor reactivity

correlating with development of an ILC1 phenotype. The direct impact of IL-35 on human NK cells and its function dampening action is a novel aspect for NK cell biology, an important finding and of interest to the field. The methodology is sound. However, all experiments are performed *in vitro* by addition of IL-35 to other cytokine combinations and the physiological role of these findings needs to be better elucidated.

We really thank the reviewer for the positive feedback on our study and we appreciate the opportunity to address the concerns and provide additional context.

While our *in vitro* experiments provide valuable insights, we recognize the importance of validating our findings *in vivo* to assess the role of IL-35 in NK cell regulation during chronic diseases. To provide new insights about the *in vivo* relevance of our *in vitro* findings, we used publicly available single-cell RNAseq datasets of human tumors (see also response below). By merging these datasets, we identify the main sources of IL-35 in human tumors and demonstrate that IL-35R+ NK cells are present in tumors and enriched in an ILC1-like gene signature. We also demonstrate the negative impact of IL-35 on clinical outcome in a TCGA pancancer analysis. These new results added as new Figure 7 provide new insights about the *in vivo* relevance of our *in vitro* findings.

We have incorporated these additional analyses and revised our manuscript to include the new data and insights gained from physiological relevance. We have revised the conclusion to acknowledge the need for additional research. Specifically, we have stated that our *in vitro* findings suggest a potential new role for IL-35 in NK cell plasticity, and that future *in vivo* studies are warranted to confirm these observations.

We believe these revisions will significantly strengthen our conclusions and address the reviewers' concerns.

• Fig 1: Since both adaptive and conventional NK cells react to IL-35, can the authors stain for the IL-35 receptor expressed on NK cells? How is the IL-35 receptor regulated?

We developed a spectral flow cytometry panel to evaluate the expression of the 3 known chains of IL-35R (gp130, IL12Rb2, and IL27Ra) in NK cells, at basal levels and upon activation by different cytokines alone (IL-12, IL-15, IL-18) or in combination 2 by 2. The gp130 chain was not expressed in resting NK cells but its expression was specifically induced by IL-12 in 40% of NK cells and the addition of IL-15 or IL-18 does not change the expression levels (Fig. 1g). Very few NK cells (5-10%) constitutively expressed IL12Rb2, while IL27Ra was expressed in 35-45% of total NK cells. IL12Rb2 was upregulated in response to individual cytokines and to a greater extent when cytokines were combined. IL27Ra is strongly upregulated in NK cells in response to IL-12, IL-15 and IL-18 and to a lesser extent to IL-12 to reach >90% IL27Ra positive NK cells in response to cytokine combination (Fig. 1g).

The expression pattern and regulation of gp130, IL12Rb2, and IL27Ra was similar between NKG2C-conventional and NKG2C+ adaptive NK cells. Furthermore, the three receptors are regulated in a similar manner upon NK cell subsets activation by cytokines used alone (IL-12, IL-15, IL-18) or in combination 2 by 2 (new Fig. 4h).

We have also used in-house transcriptomic datasets of primary NK cells that confirmed the flow cytometry data (data not included in the manuscript).

Do both subsets produce TGF- β ? and VEGF?

To answer this question, we FACS-sorted NKG2C⁺ and NKG2C⁻ NK cells, cultured them in the presence of IL-35 +/- IL-12+IL-18 for 24h, collected the culture supernatants, and quantify VEGFA and active TGF- β 1. Both NKG2C⁺ adaptive and NKG2C⁻ conventional NK cells were found to produce similar levels of active TGF- β 1 (new supplementary Figure 6d) and VEGF-A (new Figure 4j) in response to IL-35 and the addition of IL-12+IL-18 does not further increase their levels.

How is the signaling via the receptor impacting the dampening immune responses?

We examined the intracellular signaling pathways activated by IL-35 through phospho-flow cytometry but unfortunately the results were not reproducible and solid. Based on the literature, we investigated the expression of phospho-STAT1/3/4 in NK cells exposed to IL-35. However, the responsiveness of NK cells to IL-35 needs to be induced / upregulated by cytokines like IL-12, IL-15, and IL-18 that also signal through the same STAT molecules. This is a major issue that can explain the inconsistency of our results.

Does IL12 partially compete?

In the discussion of the submitted paper, we mentioned that « IL-35 inhibits IL-12 stimulation by competing with the IL-12R β 2 receptor (Mahfooz, N. S ImmunoHorizons 2023)”. This information was incomplete as the same paper also showed that IL-12 likewise disrupts IL-35 binding to IL-12R β 2 through an ELISA-like solid-state binding assay. We have modified the sentence accordingly to provide new information about the reported ability of IL-12 to compete with IL-35. In figure 1A, our results show that IL-12 does not completely abolish the inhibitory effects of IL-35, suggesting that it can partially compete with and reduce the extent of IL-35-mediated inhibition. Further research is needed to fully understand the molecular interactions of this competition.

How does IL-35 cause the delay in NK cell proliferation? Additional data should support these findings.

As shown in figure 1a, IL-35 strongly inhibited the expression of CD25/IL2R α chain which confers high affinity binding to IL-2. Thus, consequently, IL-35 may delay IL-2-triggered NK cell proliferation during the 5 day-proliferation assay. Furthermore, new experiments demonstrated that adding Galunisertib (TGF- β R inhibitor) to long-term culture in IL-2+IL-35 restored normal proliferation at similar levels as IL-2 alone, suggesting that NK proliferation is mainly inhibited by IL-35-triggered autocrine TGF- β (New Figure 6c). Our results corroborate a previous work reporting that TGF- β inhibits NK cell proliferation (Hawke J Immunol 2020 PMID 32332109).

In the scRNAseq data we actually see less NK cells in the cycling phase in the IL-2+IL-35 condition than in the IL-2 alone, but haven't checked further (see figure below).

- Fig 3: The authors show hyporesponsiveness of NK cells exposed long term to IL-35. Is this hyporesponsiveness reversible? Is NK cell produced TGF β responsible for this phenotype?

To answer to this reviewers' concern, we performed new experiments to explore whether NK hyporesponsiveness was a reversible process. NK cells were isolated from healthy donor blood, then cultured for 5 days in IL-2 +/- IL-35, IL-35-starved or not in the presence of IL-2 for 48h, and then activated with IL-12+IL-18 for 24h to assess CD25, CD9, CD103 and intracellular IFN- γ expression. We also performed the same experiments with TGF- β as control. Our results showed that IL-35-induced hyporesponsiveness was not reversible. Indeed, IL-35-starved cells produced low and similar levels of IFN- γ as NK cells that were not IL-35-starved following activation with IL-12 and IL-18 (new figure 5e and f). In contrast, we observed a partial reversion of recombinant TGF- β -induced hyporesponsiveness. Indeed, after a 48h starvation of TGF- β , culturing cells with IL-12+IL-18 led to higher production of IFN- γ compared to cells that were not TGF- β -starved (new supp figure 6b). These experiments suggest that although IL-35-mediated effects are reverted by anti-TGF- β , IL-35 has TGF- β independent activity contributing to the stabilization of their exhausted phenotype. However, we did not observe any major impact of IL-35 nor TGF- β starvation on the expression of residency markers (CD9 and CD103).

Furthermore, we demonstrated that NK cell-produced autocrine TGF- β (new Fig. 6a, Fig. 6b, and new supp Fig.6d) was responsible for the hyporesponsiveness of NK cells exposed long term to IL-35. Indeed, culturing NK cells with galunisertib (TGF- β Ri) could overcome the suppressive effect of autocrine TGF- β and allow IFN- γ production in response to IL-12+IL-18 and prevent the acquisition of a residency phenotype (CD9/CD103) (new figure 6d-f and new supp Fig. 6f).

- Since the authors claim an importance of their findings in the tumor microenvironment, it would be highly relevant to investigate the presence of IL-35 (its sources) and NK cells/ILC1 signatures in human tumors and to correlate these to tumor progression and patient survival.

By integrating four public single-cell RNAseq datasets (GSE131907, GSE160269, GSE166555, and GSE176078) into a harmonized cell atlas covering >350k cells across 108 patients (new Fig. 7a and new supp Fig. 7a-c), we analyzed the expression of IL-35 genes (*EBI3* and *IL12A*) and IL-35R chains (*IL6ST*, *IL12RB2*, *IL27RA*) in human tumors. These new results identify Breg, Treg, and to a lesser extent Dendritic cells as the main source of IL-35 in human tumors (new Fig. 7b-c and new supp Fig. 7d) and demonstrate that NK cells expressing *IL12RB2* and *IL27RA*, and to a lesser extent *IL6ST*, are present in

tumors (new Fig. 7d-e and new supp Fig. 7d) and enriched in an ILC1-like gene signature (new Fig. 7f). These new results added as new Figure 7 provide new insights about the *in vivo* relevance of our *in vitro* findings.

Furthermore, we demonstrated a positive correlation between *EBI3* and *IL12A* expression as well as between a Treg gene signature and *EBI3* or *IL12A* in a TCGA pancancer dataset (solid tumors) (new figure 7g and new supp Fig. 7e).

Finally, high expression of IL-35 (based on the coexpression of *EBI3* and *IL12A*) was associated to decreased overall survival and progression-free survival in a TCGA pancancer dataset (new figure 7h,i).

- How do the amounts of IL-35 relate to the amounts produced *in vivo*? We used 100 to 200ng/ml of IL-35 based on a dose response experiment presented in figure S1D. This is indeed much higher than amounts detected *in vivo* in plasma (100-150pg/ml in the serum of healthy donors and asthma cases, Tao P Redox Biology 2025), but consistent with the quantity of IL-35 commonly used in different *in vitro* experimental settings (Mahfooz et al, ImmunoHorizons 2023 and Li S et al, Nature 2022). This suggest high concentration of IL-35 is specific tumor niches at very close proximity of NK cells.

Minor:

- The experiments are performed with distinct cytokine combos. What happens to NK cell mediated tumor killing if NK cells are exposed to tumor cells in the context of IL-35?

In figure 3d we showed that NK cell-mediated killing of target cells (K562) was reduced if NK cells are chronically exposed to IL-35. Furthermore, our scRNAseq data revealed a decreased expression of biological pathways related to cell killing, NK cell mediated cytotoxicity, and secretory granule membrane in NK cells exposed to IL-35 compared to control NK cells (IL-2) (Figure 4d).

- The authors mention that their NK cell preparations are 95%. They should stain for the remaining cell populations and show these data.

As requested we developed a spectral flow cytometry panel to stain for the remaining cell populations in NK cell preparations. They are mainly ILC1 (<1%), ILC2 (<0.5%), and conventional DC (<0.5%). This data has been added as new supplementary Fig. 8a and see figure below for quantification, n=4 donors.

A

B

REVIEWERS' COMMENTS

Reviewer #1 (Remarks to the Author):

The authors have made significant efforts to address the reviewer's concerns and improve the manuscript. In particular, the additional analyses of IL-35-producing cells and IL-35 receptor-expressing NK cells in human tumors, along with the correlation with IL-35 levels and clinical prognosis, strengthen the significance of the study. The revised manuscript provides new insights into IL-35-mediated human NK cell regulation in tumor microenvironment and is worth publishing.

>> We thank the reviewer for recognizing the improvements we have made and for the positive feedback on our revised manuscript.

Reviewer #2 (Remarks to the Author):

My comments were addressed to my satisfaction and the data reanalysing scRNA seq of cancer patients enriched the manuscript. I noted that the legend of Fig. 7f was too small to read.

>> We thank the reviewer for recognizing the improvements we have made and for the positive feedback on our revised manuscript.

Thank you also for pointing out that the legend of Fig. 7f was too small to read. We have now increased the font size in the revised version to ensure better readability.